# Risk Minimization from Adaptively Collected Data: Guarantees for Supervised and Policy Learning

**Aurélien Bibaut**
Netflix
abibaut@netflix.com

**Nathan Kallus**
Cornell University and Netflix
kallus@cornell.edu

**Maria Dimakopoulou**
Netflix
mdimakopoulou@netflix.com

**Antoine Chambaz**
Université de Paris
antoine.chambaz@u-paris.fr

**Mark van der Laan**
University of California, Berkeley
laan@stat.berkeley.edu

## Abstract

Empirical risk minimization (ERM) is the workhorse of machine learning, whether for classification and regression or for off-policy policy learning, but its model-agnostic guarantees can fail when we use adaptively collected data, such as the result of running a contextual bandit algorithm. We study a generic importance sampling weighted ERM algorithm for using adaptively collected data to minimize the average of a loss function over a hypothesis class and provide first-of-their-kind generalization guarantees and fast convergence rates. Our results are based on a new maximal inequality that carefully leverages the importance sampling structure to obtain rates with the good dependence on the exploration rate in the data. For regression, we provide fast rates that leverage the strong convexity of squared-error loss. For policy learning, we provide regret guarantees that close an open gap in the existing literature whenever exploration decays to zero, as is the case for bandit-collected data. An empirical investigation validates our theory.

## 1 Introduction

Adaptive experiments, wherein intervention policies are continually updated as in the case of contextual bandit algorithms, offer benefits in learning efficiency and better outcomes for participants in the experiment. They also make the collected data dependent and complicate standard machine learning approaches for model-agnostic risk minimization, such as empirical risk minimization (ERM). Given a loss function and a hypothesis class, ERM seeks the hypothesis that minimizes the sample average loss. This can be used for regression, classification, and even off-policy policy optimization. An extensive literature has shown that, for independent data, ERM enjoys model-agnostic, best-in-class risk guarantees and even fast rates under certain convexity and/or margin assumptions [*e.g.* 5, 6, 35, among others]. However, these guarantees fail under contextual-bandit-collected data, both because of covariate shift due to using a context-dependent logging policy and because of the policy's data-adaptive evolution as more data are collected. A straightforward and popular approach to deal with the covariate shift is importance sampling (IS) weighting, whereby we weight samples by the inverse of the policy's probability of choosing the observed action. Unfortunately, applying standard maximal inequalities for sequentially dependent data to study guarantees of this leads to poor dependence on these weights, and therefore incorrect rates whenever exploration is decaying and the weights diverge to infinity, as happens when collecting data using a contextual bandit algorithm.

In this paper, we provide a thorough theoretical analysis of IS weighted ERM (ISWERM; pronounced "ice worm") that yields the correct rates on the convergence of excess risk under decaying exploration. To achieve this, we present a novel localized maximal inequality for IS weighted sequential empirical

35th Conference on Neural Information Processing Systems (NeurIPS 2021).

processes (Section 2) that carefully leverages their IS structure to avoid a bad dependence on the size of IS weights, as compared to applying standard results to an IS weighted process (Remark 2). We then apply this result to obtain generic slow rates for ISWERM for both Donsker-like and non-Donsker-like entropy conditions, as well as fast rates when a variance bound applies (Section 3). We instantiate these results for regression (Section 4) and for policy learning (Section 5), where we can express entropy conditions in terms of the hypothesis class and obtain variance bounds from convexity and margin assumptions. In particular, our results for policy learning close an open gap between existing lower and upper bounds in the literature (Remark 3). We end with an empirical investigation of ISWERM that sheds light on our theory (Section 6).

## 1.1 Setting

We consider data consisting of $T$ observations, $\bar{O}_T = (O_1, \dots, O_T)$, where each observation consists of a state, action, and outcome, $O_t = (X_t, A_t, Y_t) \in \mathcal{O} = \mathcal{X} \times \mathcal{A} \times \mathcal{Y}$. The spaces $\mathcal{X}, \mathcal{A}, \mathcal{Y}$ are general measurable spaces, each endowed with a base measure $\lambda_{\mathcal{X}}, \lambda_{\mathcal{A}}, \lambda_{\mathcal{Y}}$; in particular, actions can be finite or continuous (*e.g.*, $\lambda_{\mathcal{A}}$ can be counting or Lebesgue). We assume the data were generated sequentially in a stochastic-contextual-bandit fashion. Specifically, we assume that the distribution of $\bar{O}_T$ has a density $p^{(T)}$ with respect to (wrt) $\lambda_{\mathcal{O}}^T = (\lambda_{\mathcal{X}} \times \lambda_{\mathcal{A}} \times \lambda_{\mathcal{Y}})^T$, which can be decomposed as

$$p^{(T)}(\bar{o}_T) = \prod_{t=1}^T p_X(x_t) \tilde{g}_t(a_t \mid x_t, \bar{o}_{t-1}) p_Y(y_t \mid x_t, a_t),$$

where we write $\bar{o}_t = (x_1, a_1, \dots, y_t)$, using lower case for dummy values and upper case for random variables. We define $g_t(a \mid x) = \tilde{g}_t(a \mid x, \bar{O}_{t-1})$ so that $g_t$ represents the *random $\bar{O}_{t-1}$-measurable* context-dependent policy that the agent has devised at the beginning of round $t$, which they then proceed to employ when observing $X_t$. Since we run the adaptive experiment, we assume that $g_t$ is known, as it is actually computed at stage $t$ of the experiment. We do *not* assume that $\tilde{g}_t$ is known.

**Remark 1** (Counterfactual interpretation). We can also interpret this data collection from a counterfactual perspective. At the beginning of each round, $(X_t, \{Y_t(a) : a \in \mathcal{A}\})$ is drawn from some stationary (*i.e.*, time-independent) distribution $P^*$, $X_t$ is revealed, and after acting with a non-anticipatory action $A_t$, meaning it only depends on past data, we observe $Y_t = Y_t(A_t)$. This corresponds to the above with $p_X$ being the marginal of $X_t$ under $P^*$ and $p_Y(\cdot \mid x, a)$ the conditional distribution of $Y_t(a)$ given $X_t = x$.

## 1.2 Importance Sampling Weighted Empirical Risk Minimization

Consider a class of hypotheses $\mathcal{F}$, a loss function $\ell : \mathcal{F} \times \mathcal{O} \to \mathbb{R}$, and some fixed reference $g^*(a \mid x)$, any function, for example, a conditional density. As we will see in Examples 1 to 3 below we will often simply use $g^*(a \mid x) = 1$. Define the *population reference risk* as

$$R^*(f) = \mathbb{E}_{p_X \times g^* \times p_Y}[\ell(f, O)] = \int \ell(f, (x, a, y)) p_Y(y \mid x, a) g^*(a \mid x) p_X(x) d\lambda_{\mathcal{O}}(x, a, y).$$

We are interested in finding $f$ with low risk $R^*(f)$. We consider doing so using ISWERM, which is ERM where we weight each term by the density ratio between the reference and the policy at time $t$:

$$\hat{f}_T \in \operatorname*{argmin}_{f \in \mathcal{F}} \left\{ \hat{R}_T(f) = \frac{1}{T} \sum_{t=1}^T \frac{g^*(A_t \mid X_t)}{g_t(A_t \mid X_t)} \ell(f, O_t) \right\}.$$

**Example 1** (Regression). Consider $\mathcal{Y} = \mathbb{R}$, $\mathcal{F} \subseteq [\mathcal{X} \times \mathcal{A} \to \mathbb{R}]$, and $\ell(f, o) = (y - f(x, a))^2$. Then $f$ with small $R^*(f)$ is good at predicting outcomes from context and action. In particular, for any $g^*$, we have that $\mu(x, a) = \int y p_Y(y \mid x, a) d\lambda_Y(y)$ solves $\mu \in \operatorname{argmin}_{f: \mathcal{X} \times \mathcal{A} \to \mathcal{Y}} R^*(f)$. And, we can write $R^*(f) - R^*(\mu) = \mathbb{E}_{p_X \times g^*}[(f - \mu)^2(X, A)]$.

Consider the counterfactual interpretation in Remark 1. Then $R^*(f) = \int \mathbb{E}_{P^*}[(Y(a) - f(X, a))^2] g^*(x \mid a) d\lambda_{\mathcal{A}}(a)$. For example, if $|\mathcal{A}| < \infty$, $\lambda_{\mathcal{A}}$ is the counting measure, and $g^*(x \mid a) = 1$, then $R^*(f) = \sum_{a \in \mathcal{A}} \mathbb{E}_{P^*}[(Y(a) - f(X, a))^2]$ is the total counterfactual prediction error. Alternatively, if $g^*(a \mid x) = \mathbf{1}(a = a^*)$ and given some $\mathcal{H} \subseteq [\mathcal{X} \to \mathcal{Y}]$ we let $\mathcal{F} = \{f_h(x, a) = h(x) : h \in \mathcal{H}\}$, then we have $R^*(f_h) = \mathbb{E}_{P^*}[(Y(a^*) - h(X))^2]$, that is, the regression risk for predicting the counterfactual outcome $Y(a^*)$ from $X$.

**Example 2** (Classification). In the same setting as Example 1, suppose $\mathcal{Y} = \{\pm 1\}$. Then $\mu(x, a) = 2p_Y(1 \mid x, a) - 1$. And, if we restrict $\mathcal{F} \subseteq [\mathcal{X} \times \mathcal{A} \to \{\pm 1\}]$, letting $\ell(f, o) = \frac{1}{2} - \frac{1}{2}yf(x, a)$ leads to $R^*(f)$ being misclassification rate, an unrestricted minimizer of which is $\text{sign}(\mu(x, a))$. Focusing on misclassification of $\text{sign}(f(x, a))$ for $\mathcal{F} \subseteq [\mathcal{X} \times \mathcal{A} \to \mathbb{R}]$, we can also use a classification-calibrated loss [6], such as logistic $\ell(f, o) = \log(1 + \exp(-yf(x, a)))$, hinge $\ell(f, x) = (1 - yf(x, a))_+$, *etc.*.

**Example 3** (Policy learning). Consider $\mathcal{Y} = \mathbb{R}$, $\mathcal{F} \subseteq [\mathcal{X} \times \mathcal{A} \to \mathbb{R}]$, $g^*(a \mid x) = 1$ and $\ell(f, o) = yf(x, a)$. Then $R^*(f) = \int yp_Y(y \mid x, a)d\lambda_Y(y)f(a \mid x)d\lambda_A(a)p_X(x)d\lambda_X(x) = \mathbb{E}_{p_X \times f \times p_Y}[y]$ is the average outcome under a policy $f$. If we interpret outcomes as costs (or, negative rewards), then seeking to minimize $R^*(f)$ means to seek a policy with least risk (or, highest value).

Consider in particular the counterfactual interpretation in Remark 1 with $|\mathcal{A}| < \infty$. Consider deterministic policies: given $\mathcal{H} \subseteq [\mathcal{X} \to \mathcal{A}]$, let $\mathcal{F} = \{f_h(x, a) = \mathbf{1}(h(x) = a) : h \in \mathcal{H}\}$. Then we have $R^*(f_h) = \mathbb{E}_{P^*}[Y(h(X))]$, that is, the average counterfactual outcome.

## 1.3 Related Literature

**Contextual bandits.** A rich literature studies how to design adaptive experiments to optimize regret, simple regret, or the chance of identifying best interventions [see 12, 36, and bibliographies therein]. Such adaptive experiments can significantly improve upon randomized trials (aka A/B tests), which is why they are seeing increased use in practice in a variety of settings, from e-commerce to policymaking [3, 4, 29, 32, 37, 43, 44, 53]. However, while randomized trials produce iid data, adaptive experiments do not, complicating post-experiment analysis, which motivates our current study. Many stochastic contextual bandit algorithms (stochastic meaning the context and response models are stationary, as in our setting) need to tackle learning from adaptively collected data to fit regression estimates of mean reward functions, but for the most part this is based on models such as linear [7, 14, 23, 37] or Hölder class [26, 47, 47], rather than on doing model-agnostic risk minimization and nonparametric learning with general function classes as we do here. Foster and Rakhlin [19] use generic regression models but require online oracles with guarantees for adversarial sequences of data. Simchi-Levi and Xu [50] use offline least-squares ERM but bypass the issue of adaptivity by using epochs of geometrically growing size in each of which data are collected iid. Other stochastic contextual bandit algorithms are based on direct policy learning using ERM [1, 8, 16, 20]; by carefully designing exploration strategies, they obtain good regret rates that are even better than the minimax-optimal guarantees given only the exploration rates, as we obtain (Remark 3).

**Inference with adaptive data.** A stream of recent literature tackles how to construct confidence intervals after an adaptive experiment. While standard estimators like inverse-propensity weighting (IPW) and doubly robust estimation remain unbiased under adaptive data collection, they may no longer be asymptotically normal making inference difficult. To fix this, Hadad et al. [24] use and generalize a stabilization trick originally developed by Luedtke and van der Laan [38] for a non-adaptive setting with different inferential challenges. Their stabilized estimator, however, only works for data collected by non-contextual bandits. Bibaut et al. [9] extend this to a contextual-bandit setting. Our focus is different from these: risk minimization and guarantees rather than inference.

**Policy learning with adaptive data.** Zhan et al. [58] study policy learning from contextual-bandit data by optimizing a doubly robust policy value estimator stabilized by a deterministic lower bound on IS weights. They provide regret guarantees for this algorithm based on invoking the results of Rakhlin et al. [45]. However, these guarantees do not match the algorithm-agnostic lower bound they provide whenever the lower bounds on IS weights decay to zero, as they do when data are generated by a bandit algorithm. For example, for an epsilon-greedy bandit algorithm with an exploration rate of $\epsilon_t = t^{-\beta}$, their lower bound on expected regret is $\Omega(T^{-(1-\beta)/2})$ while their upper bound is $O(T^{-(1/2-\beta)})$. We close this gap by providing an upper bound of $O(T^{-(1-\beta)/2})$ for our simpler IS weighted algorithm. See Remark 3. Our results for policy learning also extend to fast rates under margin conditions, non-Donsker-like policy classes, and learning via convex surrogate losses.

**IS weighted ERM.** The use of IS weighting to deal with covariate shift, including when induced by a covariate-dependent policy, is standard. For estimation of causal effects from observational data this usually takes the form of inverse propensity weighting [28]. The same is often used for ERM for regression [15, 22, 48] and for policy learning [34, 52, 59]. When regressions are plugged into causal effect estimators, weighted regression with weights that depend on IS weights minimize the resulting estimation variance over a hypothesis class [13, 18, 30, 31, 49]. All of these approaches however

have been studied in the independent-data setting where historical logging policies do not depend on the same observed data available for training, guarantees under which is precisely our focus herein.

**Sequential maximal inequalities.** There are essentially two strands in the literature on maximal inequalities for sequential empirical processes. One expresses bounds in terms of sequential bracketing numbers as introduced by van de Geer [55], generalizing of standard bracketing numbers. Another uses sequential covering numbers, introduced by Rakhlin et al. [46]. These are in general not comparable. Foster and Krishnamurthy [20], Zhan et al. [58] use sequential $L_\infty$ and $L_p$ covering numbers, respectively, to obtain maximal inequalities. van de Geer [55, Chapter 8] gives guarantees for ERM over nonparametric classes of controlled sequential bracketing entropy. However, applying her generic result as-is to IS weighted processes provides bad dependence on the exploration rate in the case of larger-than-Donsker hypothesis classes (see Remark 2). We also use sequential bracketing numbers, but we develop a new maximal inequality specially for IS weighted sequential empirical processes, where we use the special structure when truncating the chaining to avoid a bad dependence on the size of the IS weights. Equipped with our new maximal inequality, we obtain first-of-their kind guarantees for ISWERM, including fast rates that have not been before derived in adaptive settings.

## 2 A Maximal Inequality for IS Weighted Sequential Empirical Processes

A key building block for our results is a novel maximal inequality for IS weighted sequential empirical processes. For any sequence of objects $(x_t)_{t \geq 1}$, we introduce the shorthand $x_{1:T}$ to denote the sequence $(x_t)_{t=1}^T$. We say that a sequence of random variables $\zeta_{1:T}$ is $\bar{O}_{1:T}$-predictable if, for every $t \in [T] = \{1, \ldots, T\}$, $\zeta_t$ is $\bar{O}_{t-1}$-measurable, *i.e.*, is some function of $\bar{O}_{t-1}$.

**IS weighted sequential empirical processes.** Let $P_g$ denote the distribution on $\mathcal{O} = \mathcal{X} \times \mathcal{A} \times \mathcal{Y}$ with density w.r.t. $\lambda_X \times \lambda_A \times \lambda_Y$ given by $p_X \times g \times p_Y$ and let us use the notation $P_g h(o) := \int h(o) dP_g(o)$. Let us also define the norm $\|h\|_{p,g} = (P_g(h^p))^{1/p}$. Consider a sequence of $\mathcal{F}$-indexed random processes of the form $\Xi_T := \{(\xi_t(f))_{t=1}^T : f \in \mathcal{F}\}$ where, for every $f \in \mathcal{F}$, $\xi_{1:T}(f)$ is an $\bar{O}_{1:T}$-predictable sequence of $\mathcal{O} \to \mathbb{R}$ functions. The IS-weighted sequential empirical process induced by $\Xi_T$ is the $\mathcal{F}$-indexed random process

$$M_T(f) := \frac{1}{T} \sum_{t=1}^T \frac{g_t^*(A_t \mid X_t)}{g_t(A_t \mid X_t)} \left( \xi_t(f)(O_t) - \mathbb{E}\left[ \xi_t(f)(O_t) \mid \bar{O}_{t-1} \right] \right)$$

$$= \frac{1}{T} \sum_{t=1}^T (\delta_{O_t} - P_{g_t}) \left( \frac{g^*}{g_t} \xi_t(f) \right).$$

**Sequential bracketing entropy.** For any $\bar{O}_{1:T}$-predictable sequence sequence $\zeta_{1:T}$ of functions $\mathcal{O} \to \mathbb{R}$, we introduce the pseudonorm $\rho_{T,g^*}(\zeta_{1:T}) := (T^{-1} \sum_{t=1}^T \|\zeta_t\|_{2,g^*}^2)^{1/2}$.

We say that a collection of $2N$-many $\bar{O}_{1:T}$-predictable sequences of $\mathcal{O} \to \mathbb{R}$ functions $\{(\lambda_{1:T}^{(k)}, \upsilon_{1:T}^{(k)}) : k \in [N]\}$ is an $(\epsilon, \rho_{T,g^*})$-sequential bracketing of $\Xi_T$, if (a) for every $f \in \mathcal{F}$, there exists $k \in [N]$ such that $\lambda_t^{(k)} \leq \xi_t(f) \leq \upsilon_t^{(k)} \ \forall t \in [T]$ and (b) for every $k \in [N]$, $\rho_{T,g^*}(\upsilon_{1:T}^{(k)} - \lambda_{1:T}^{(k)}) \leq \epsilon$. We denote by $\mathcal{N}_{[]}(\epsilon, \Xi_T, \rho_{T,g^*})$ the minimal cardinality of an $(\epsilon, \rho_{T,g^*})$-sequential bracketing of $\Xi_T$.

**The special case of classes of classes of deterministic functions.** Consider the special case $\xi_t(f) := \xi(f)$, where $\Xi := \{\xi(f) : f \in \mathcal{F}\}$ is a class of functions where for every $f \in \mathcal{F}$, $\xi(f)$ is a deterministic $\mathcal{O} \to \mathbb{R}$ function. Observe that for a fixed function $\zeta : \mathcal{O} \to \mathbb{R}$, letting $\zeta_t := \zeta$, we have that $\rho_{T,g^*}(\zeta_{1:T}) = \|\zeta\|_{2,g^*}$. Therefore, $\mathcal{N}_{[]}(\epsilon, \Xi_T, \rho_{T,g^*})$, the $(\epsilon, \rho_{T,g^*})$-sequential bracketing number of $\Xi_T$, reduces to $N_{[]}(\epsilon, \Xi, \|\cdot\|_{2,g^*})$, the usual $\epsilon$-bracketing number $\Xi$ in the $\|\cdot\|_{2,g^*}$ norm.

**The maximal inequality.** Our maximal inequality will crucially depend on the decay rate of the the IS weights, that is, the exploration rate of the adaptive data collection.

**Assumption 1.** There exists a deterministic sequence of positive numbers $(\gamma_t)$ such that, for any $t \geq 1$, $\|g^*/g_t\|_\infty \leq \gamma_t$, almost surely. Define $\gamma_T^{\mathrm{avg}} := T^{-1} \sum_{t=1}^T \gamma_t$ and $\gamma_T^{\mathrm{max}} := \max_{t \in [T]} \gamma_t$.

For example, if the data were collected under an $\epsilon_t$-greedy contextual bandit algorithm then we have $\gamma_t = \epsilon_t^{-1}$. If we have $\epsilon_t = t^{-\beta}$ for $\beta \in (0, 1)$ then $\gamma_T^{\mathrm{max}} = O(\gamma_T^{\mathrm{avg}}) = O(T^\beta)$. Note that

having $\gamma_t < \infty$ in Assumption 1 does restrict us to contextual bandit algorithms that are conditionally random, such as $\epsilon$-greedy and Thompson sampling, and rules out algorithms with deterministic policies given the history, such as UCB.

**Theorem 1.** *Consider* $\Xi_T := \{\xi_{1:t}(f) : f \in \mathcal{F}\}$ *as defined above. Suppose that Assumption 1 holds, and that there exists $B > 0$ such that $\max_{t \in [T]} \sup_{f \in \mathcal{F}} \|\xi_t(f)\|_\infty \leq B$. In the special case where $\xi_t(f) = \xi(f)$, $\xi_{1:T} \in \Xi_T$, are deterministic functions, we let $\widetilde{\gamma}_T := \gamma_T^{avg}$. Otherwise, in the general case, we let $\widetilde{\gamma}_T := \gamma_T^{\max}$. Let $r > 0$. Let $\mathcal{F}_T(r) := \{f \in \mathcal{F} : \rho_{T,g^*}(\xi_{1:T}(f)) \leq r\}$. For any $r^- \in [0, r/2]$, and any $x > 0$, it holds with probability at least $1 - 2e^{-x}$ that*

$$\sup_{f \in \mathcal{F}_T(r)} M_T(f) \lesssim r^- + \sqrt{\frac{\widetilde{\gamma}_T}{T}} \int_{r^-}^r \sqrt{\log(1 + \mathcal{N}_{[]}(\epsilon, \Xi_T, \rho_{T,g^*}))} d\epsilon$$

$$+ \frac{\gamma_T^{\max} B}{T} \log(1 + \mathcal{N}_{[]}(r, \Xi_T, \rho_{T,g^*})) + \sqrt{\frac{\widetilde{\gamma}_T x}{T}} + \frac{B \gamma_T^{\max} x}{T}.$$

**Remark 2** (Leveraging IS structure)**.** Theorem 1 is based on a finite-depth adaptive chaining device, in which we leverage the IS-weighted structure to carefully bound the size of the tip of the chains. In contrast, applying Theorem 8.13 of van de Geer [55] to the IS weighted sequential empirical process would lead to suboptimal dependence on $\gamma_t$. The crucial point is to work with IS-weighted chains of the form $(g^*/g_t)\xi(f) = (g^*/g_t)\{(\xi(f) - u^{J,f}) + \sum_{j=0}^J (u^{j,f} - u^{j-1,f}) + u^{0,f}\}$, where the $u^{j,f}$ are upper brackets of the unweighted class $\Xi$, at scales $\epsilon_1 > \ldots > \epsilon_J$ (we simplify here a bit the chaining decomposition for ease of presentation compared with the proof). In adaptive chaining, the tip is bounded by the $L_1$ norm of the corresponding bracket. In our case, denoting $l^{J,f}$ the lower bracket corresponding to $u^{J,f}$, the tip is bounded by $P_{g_t}(g^*/g_t)|u^{J,f} - l^{J,f}| = P_{g^*}|u^{J,f} - l^{J,f}|$, in which we integrate out the IS ratio, thereby paying no price for it. Applying directly Theorem 8.13 of van de Geer [55], we would be working with a bracketing of the IS weighted class $\{(g^*/g_t)\xi(f) : f \in \mathcal{F}\}$. When working with generic $L_2$ brackets of the weighted class, the IS-weighting structure is lost, and we cannot do better than bounding the $L_1$ of the tip by its $L_2$ norm, which depends on $\gamma_t$. Since in sequential settings, $\gamma_t$ generally diverges to $\infty$, good dependence is paramount to obtaining tight, informative results. Our proof technique otherwise follows the same general outlines as those of van de Geer [55, Theorem 8.13] and van Handel [57, Theorem A.4] (or, 40, Theorem 6.8 in the iid setting). Like these, we too leverage an adaptive chaining device, as pioneered by Ossiander [41].

## 3  Applications to Guarantees for ISWERM

We now return to ISWERM and use Theorem 1 to obtain generic guarantees for ISWERM. We will start with so-called slow rates that give generic generalization results and then present so-called fast rates that will apply in certain settings, where a so-called variance bound is available. Let $f_1$ be a minimizer of the population risk $R^*$ over $\mathcal{F}$, that is $f_1 \in \arg\min_{f \in \mathcal{F}} R^*(f)$.

**Assumption 2** (Entropy on $\ell(\mathcal{F})$)**.** Define $\ell(\mathcal{F}) := \{\ell(f, \cdot) : f \in \mathcal{F}\}$. There exist an envelope function $\Lambda : \mathcal{O} \to \mathbb{R}$ of $\ell(\mathcal{F})$, and $p > 0$ such that, for any $\epsilon > 0$,

$$\log N_{[]}(\epsilon \|\Lambda\|_{2,g^*}, \ell(\mathcal{F}), \|\cdot\|_{2,g^*}) \lesssim \epsilon^{-p}.$$

The case $p < 2$ corresponds to the Donsker case, and $p \geq 2$ to the (possibly) non-Donsker case.

**Assumption 3** (Diameters on $\ell(\mathcal{F})$)**.** There exist $b_0 > 0$ and $\rho_0 > 0$ such that

$$\sup_{f \in \mathcal{F}} \|\ell(f, \cdot) - \ell(f_1, \cdot)\|_\infty \leq b_0 \|\Lambda\|_{2,g^*}, \qquad \sup_{f \in \mathcal{F}} \|\ell(f, \cdot) - \ell(f_1, \cdot)\|_{2,g^*} \leq \rho_0 \|\Lambda\|_{2,g^*}.$$

**Theorem 2** (Slow Rates for ISWERM)**.** *Suppose Assumptions 1 to 3 hold. Then for any $\delta \in (0, 1/2)$, we have that, with probability at least $1 - \delta$,*

$$R^*(\hat{f}_T) - \inf_{f \in \mathcal{F}} R^*(f)$$

$$\lesssim \|\Lambda\|_{2,g^*} \times \begin{cases} \rho_0 \sqrt{\frac{\gamma_T^{avg}}{T}} \left\{\rho_0^{-p/2} + \sqrt{\log(1/\delta)}\right\} + \frac{b_0 \gamma_T^{\max}}{T} \left\{\rho_0^{-p} + \log(1/\delta)\right\} & p < 2, \\ \left(\frac{\gamma_T^{avg}}{T}\right)^{\frac{1}{p}} + \rho_0 \sqrt{\frac{\gamma_T^{avg}}{T}} \sqrt{\log(1/\delta)} + \frac{b_0 \gamma_T^{\max}}{T} \left\{\rho_0^{-p} + \log(1/\delta)\right\} & p > 2. \end{cases}$$

For $p = 2$ the bound is similar to the second case but with polylog terms; for brevity we omit the $p = 2$ case in this paper. Theorem 2 suggests that the excess risk of ISWERM converges at the rate of $(\gamma_T^{\mathrm{avg}}/T)^{\frac{1}{p} \wedge \frac{1}{2}}$. For example, if $\gamma_T^{\mathrm{avg}} = O(T^\beta)$ and $p < 2$, we obtain $O(T^{-\frac{1}{2}(1-\beta)})$. For $\beta = 0$ this matches the familiar slow rate of iid settings. However, in many cases we can obtain faster rates.

**Assumption 4** (Variance Bound). For some $\alpha \in (0, 1]$, we have

$$\|\ell(f, \cdot) - \ell(f_1, \cdot)\|_{2, g^*} \lesssim \|\Lambda\|_{2, g^*} \left( \frac{R^*(f) - R^*(f_1)}{\|\Lambda\|_{2, g^*}} \right)^{\frac{\alpha}{2}} \quad \forall f \in \mathcal{F}.$$

As we will see in Lemmas 2 and 3, we can ensure Assumption 4 holds for least-squares regression and for policy learning with a margin condition.

**Assumption 5** (Convexity). $\mathcal{F}$ is convex and $\ell(\cdot, O)$ is almost surely convex.

**Theorem 3** (Fast Rates for ISWERM). *Suppose Assumptions 1 to 5 hold with $p < 2$. Then for any $\delta \in (0, 1/2)$, we have that, with probability at least $1 - \delta$,*

$$R^*(\hat{f}_T) - R^*(f_1) \lesssim \|\Lambda\|_{2, g^*} \times \left\{ \left( \frac{\gamma_T^{avg}}{T} \right)^{\frac{1}{2 - \alpha + p\alpha/2}} + \left( \frac{b_0 \gamma_T^{\max}}{T} \right)^{\frac{1}{1 + p\alpha/2}} \right.$$

$$\left. + \left( \frac{\gamma_T^{avg} \log(1/\delta)}{T} \right)^{\frac{1}{2 - \alpha}} + \frac{b_0 \gamma_T^{\max} \log(1/\delta)}{T} \right\}$$

**The entropy condition.** Assumption 2 assumes an entropy bound on the loss class $\ell(\mathcal{F})$. For many loss functions, we can easily satisfy this condition by assuming an entropy condition on $\mathcal{F}$ itself.

**Assumption 6** (Entropy on $\mathcal{F}$). There exists $p > 0$ and an envelope function $F$ of $\mathcal{F}$ such that

$$\log N_{[]}(\epsilon \|F\|_{2, g^*}, \mathcal{F}, \| \cdot \|_{2, g^*}) \lesssim \epsilon^{-p}.$$

**Lemma 1** (Lemma 4 in Bibaut and van der Laan [10]). *Suppose that $\{\ell(\cdot, o) : o \in \mathcal{O}\}$ is a set of $\mathbb{R} \to \mathbb{R}$ unimodal functions that are equi-Lipschitz. Then Assumption 6 implies Assumption 2.*

There are many examples of $\mathcal{F}$ for which bracketing entropy conditions are known. The class of $\beta$-Hölder smooth functions (meaning having derivatives of orders up to $\mathfrak{b} = \sup\{i \in \mathbb{Z} : i < \beta\}$ and the $\mathfrak{b}$-order derivatives are $(\beta - \mathfrak{b})$-Hölder continuous) on a compact domain in $\mathbb{R}d$ has $p = d/\beta$ [56, Corollary 2.7.2]. The class of *convex* Lipschitz functions on a compact domain in $\mathbb{R}d$ has $p = d/2$ [56, Corollary 2.7.10]. The class of monotone functions on $\mathbb{R}$ has $p = 1$ [56, Theorem 2.7.5]. If $\mathcal{F} = \{f(o; \theta) : \theta \in \Theta\}$, $f(o; \theta)$ is Lipschitz in $\theta$, and $\Theta \subseteq \mathbb{R}d$ is compact, then any $p > 0$ holds [56, Theorem 2.7.11]. The class of càdlàg functions $[0, 1]^d \to \mathbb{R}$ with sectional variation norm (aka Hardy-Krause variation) no larger than $M > 0$ has envelope-scaled bracketing entropy $O(\epsilon^{-1} |\log(1/\epsilon)|^{2(d-1)})$ [10], so Assumption 6 holds with any $p > 1$ (or, we can track the log terms). Since trees with bounded output range and finite depth fall in the class of càdlàg functions with bounded sectional variation norm, decision tree classes also satisfy Assumption 6 with any $p > 1$.

## 4 Least squares regression using ISWERM

We now instantiate ISWERM for least squares regression. Consider $\mathcal{Y} = [-\sqrt{M}, \sqrt{M}]$, for some $M > 0$, $\mathcal{F} \subseteq [\mathcal{X} \times \mathcal{A} \to \mathcal{Y}]$, and $\ell(f, o) = (y - f(x, a))^2$. If $\mathcal{F}$ is convex, strongly convex losses such as $\ell$ always yield a variance bound with respect to any population risk minimizer over $\mathcal{F}$ (see e.g. lemma 15 in [6]). Let $f_1 \in \operatorname{argmin}_{f \in \mathcal{F}} R^*(f)$ be such a population risk minimizer. We present in the lemma below properties relevant for application of theorems 2 and 3

**Lemma 2** (Properties of the square loss.). *Consider the setting of the current section. The square loss $\ell$ over $\mathcal{F} \times \mathcal{O}$ satisfies the following variance bound:*

$$\|\ell(f, \cdot) - \ell(f_1, \cdot)\|_{2, g^*} \leq 4\sqrt{M}(R^*(f) - R^*(f_1))^{1/2} \, \forall f \in \mathcal{F},$$

*and the following Lipschitz property:*

$$|\ell(f, o) - \ell(f', o)| \leq \sqrt{M}|f(a, x) - f'(a, x)| \, \forall f, f' \in \mathcal{F}, o \in \mathcal{O}.$$

**Theorem 4** (Least squares regression). *Suppose Assumption 1 holds. Suppose Assumption 6 holds for the envelope taken to be constant equal to $\sqrt{M}$, the range of the regression functions. Then for any $\delta \in (0, 1/2)$, we have that, with probability at least $1 - \delta$,*

$$R^*(\widehat{f}_T) - R^*(f_1) \lesssim M \times \begin{cases} \left(\frac{\gamma_T^{\max}}{T}\right)^{\frac{1}{1+p/2}} + \frac{\gamma_T^{\max}\log(1/\delta)}{T} & \text{if } p < 2, \\ \left(\frac{\gamma_T^{avg}}{T}\right)^{\frac{1}{p}} + \frac{\gamma_T^{\max}}{T} + \sqrt{\frac{\gamma_T^{avg}\log(1/\delta)}{T}} + \frac{\gamma_T^{\max}\log(1/\delta)}{T} & \text{if } p > 2. \end{cases}$$

## 5 Policy Learning using ISWERM

We next instantiate ISWERM for policy learning. Consider $\mathcal{Y} = [-M, M]$, $\mathcal{F} \subseteq [\mathcal{X} \times \mathcal{A} \to \mathbb{R}]$ as in Example 3. Let $\ell(f, o) = yf(x, a)$ and $g^*(a \mid x) = 1$ so that $P_{g^*}\ell(f, \cdot) = \mathbb{E}_{p_X \times f \times p_Y}[y] = \mathbb{E}_{p_X}[\sum_{a \in \mathcal{A}} f(X, a)\mu(X, a)]$ is exactly the average outcome under a policy $f$ (or, its negative value), where we define $\mu(x, a) = \int y p_Y(y \mid x, a) d\lambda_Y(y)$.

We first give specification-agnostic slow rates, which also close an open gap in the literature.

**Theorem 5** (ISWERM Policy Learning: slow rates). *Suppose Assumption 1 holds and suppose that Assumption 6 holds withe envelope constant equal to 1 (which is the maximal range of policies). Then for any $\delta \in (0, 1/2)$, we have that, with probability at least $1 - \delta$,*

$$R^*(\hat{f}_T) - \inf_{f \in \mathcal{F}} R^*(f) \lesssim M \times \begin{cases} \sqrt{\frac{\gamma_T^{avg}}{T}}\sqrt{\log(1/\delta)} + \frac{\gamma_T^{\max}}{T}\log(1/\delta) & p < 2, \\ \left(\frac{\gamma_T^{avg}}{T}\right)^{\frac{1}{p}} + \sqrt{\frac{\gamma_T^{avg}}{T}}\sqrt{\log(1/\delta)} + \frac{\gamma_T^{\max}}{T}\log(1/\delta) & p > 2. \end{cases}$$

**Remark 3** (Comparison to Zhan et al. [58]). Given a deterministic policy class ($f_h(x, a) = \mathbb{1}(h(x) = a)$), $\mathcal{H} \subseteq [\mathcal{X} \to \mathcal{A}]$) with a Natarajan dimension, Zhan et al. [58] show a lower bound of $\Omega((\gamma_T^{avg}T)^{-1/2})$ on the expected regret of any policy-learning algorithms for some logging policy satisfying Assumption 1 (see their Theorem 1), that is, $\Omega(T^{-(1-\beta)/2})$ when $\gamma_T^{avg} = \Omega(T^\beta)$. Zhan et al. [58] also provide an upper bound of $O(\gamma_T^{avg}T^{-1/2})$ for their particular algorithm (see their Corollary 2.1), that is, $O(T^{-1/2+\beta})$ when $\gamma_T^{avg} = O(T^\beta)$, assuming that $\log N_H(\epsilon, \mathcal{H}) \lesssim \epsilon^{-p}$ for $p < 1$, where $N_H$ is the Hamming covering number, which would be implied by having a Natarajan dimension. This is a potentially significant gap in the regret rate in $T$ when exploration is diminishing, $\beta > 0$, as is often the case with bandit-collected data. For example, for $\beta = 1/2$, this yields a vanishing lower bound and a non-vanishing upper bound.

In comparison, our Theorem 5 gives the rate $O((\gamma_T^{avg}T)^{-1/2})$ on the expected regret of policy learning with ISWERM in the case of $p < 2$ (given by integrating the tail inequality in Theorem 5), that is, $O(T^{-(1-\beta)/2})$ when $\gamma_T^{avg} = O(T^\beta)$. This matches the rate of the lower bound of Zhan et al. [58], seemingly closing the gap. While Natarajan dimension, Hamming covering entropy with $p < 1$, and bracketing entropy with $p < 2$ are generally incomparable conditions (aside from the first implying the second), they all generally hold for policy classes parametrized by finite-dimensional parameters and tree policy classes, for which we definitely close gap. It remains an open question how to close the gap for general policy classes that only satisfy a Hamming covering entropy condition.

The gap arose from the specific technical route Zhan et al. [58] followed (not their algorithm). For the sake of exposition, we give an explanation of the phenomenon in a non-sequential i.i.d. setting, under stationary logging policy $g_1$, and under our own notation. The same phenomenon translates to the sequential setting. Since they use a symmetrization and covering-based approach, they need to work with uniform covering-type entropies[1] of the form $\sup_Q \log N(\epsilon, \mathcal{H}, L_2(Q))$ for a certain class $\mathcal{H}$, where the supremum is over all finitely supported distributions. Their approach amounts to taking $\mathcal{H}$ to be the weighted loss class $\{(g^*/g_t)\ell(\pi)\}$. While for $Q = P_{g_1}$, it holds that $\|(g^*/g_1)(\ell(\pi) - \ell(\pi'))\|_{2,P_{g_1}} \leq \gamma_1^{1/2}\|\ell(\pi) - \ell(\pi')\|_{2,P_{g^*}} \lesssim \gamma_1^{1/2}d_H(\pi, \pi')$, for a general $Q$, the best bound is $\|(g^*/g_1)(\ell(\pi) - \ell(\pi'))\|_{2,Q} \lesssim \gamma_1 d_H(\pi, \pi')$, where $d_H$ is the Hamming distance. By contrast, when working with bracketing entropy, one only needs to control the size of brackets in terms of $L_2(P_{g_1})$, that is the $L_2$-norm under the distribution of the data $P_{g_1}$. This allows to save a

---

[1]See van der Vaart and Wellner [56, Chapter 2.3] for an explanation of why uniform covering entropy is natural for bounding symmetrized Rademacher processes.

$\sqrt{\gamma_1}$ factor. Our results also show that a simple IS weighting algorithm suffices to obtain optimal rates, and the stabilization by $\gamma_t$ employed by Zhan et al. [58], which is inspired by the stabilization employed by Hadad et al. [24], Luedtke and van der Laan [38] for inference purposes, may not be necessary for policy learning purposes. The doubly-robust-style centering may still be beneficial in practice for reducing variance but it does not affect the rate.

**Remark 4** (Comparison to [20]). Foster and Krishnamurthy [20], albeit in a slightly different setting, derive a maximal inequality under sequential covering entropy that also exhibits the correct dependence on the exploration rate as ours. This shows in particular that the suboptimal dependence on the exploration rate of Zhan et al. [58] is not a necessary consequence of using sequential covering entropy. Analogously to us, Foster and Krishnamurthy [20] exploit the specific IS-weigthed structure of the loss process, and work with covers of the unweighted policy class directly. Using an $L_\infty$ sequential cover of the unweighted class and using Holder's inequality, they are able to factor out the $L_1$ norm of the IS ratios. This allows them to circumvent the type of sequential cover of the weighted class that Zhan et al. [58] need, and yields optimal $\gamma$ scaling. One caveat of this approach is that the entropy integral in the corresponding bound is expressed in terms of $L_\infty$ sequential covering entropy, which makes it hard to obtain fast rates via localization. Indeed, while variance bounds that allow for localization in $L_2$ norm are common, it is in general much harder to obtain localization in $L_\infty$ norm.

In well-specified cases, much faster rates of regret convergence are possible. We focus on finitely-many actions, $|\mathcal{A}| < \infty$. Define $\mu^*(X) = \min_{a \in \mathcal{A}} \mu(X, a)$ and fix $a^*(X)$ with $\mu(X, a^*(X)) = \mu^*(X)$.

**Assumption 7** (Margin). For a constant $\nu \in [0, \infty]$, we have for all $u \geq 0$,

$$\Pr_{p_X} \left( \min_{a \in \mathcal{A} \setminus \{a^*(X)\}} \mu(X, a) - \mu^*(X) \leq Mu \right)^{1/\nu} \lesssim u,$$

where we define $0^{1/\infty} = 0$ and $x^{1/\infty} = 1$ for $x \in (0, 1]$.

This type of margin condition was originally considered in the case of binary classification [39, 54]. The condition we use is more similar to that used in multi-arm contextual bandits [25, 26, 42]. The condition controls the density of the arm gap near zero. It generally holds with $\nu = 1$ for sufficiently well-behaved $\mu$ and continuous $X$ and with $\nu = \infty$ for discrete $X$ [see, *e.g.*, 27, Lemmas 4 and 5].

**Lemma 3.** *Suppose Assumption 7 holds and $\min_{f \in \mathcal{F}} R^*(f) = \mathbb{E}_{p_X} \mu^*(X)$. Then Assumption 4 holds for $\alpha = \nu/(\nu + 1)$ and $\Lambda : o \mapsto M$.*

**Theorem 6** (ISWERM Policy Learning: fast rates). *Suppose Assumptions 1, 6 and 7 hold with $p < 2$ and $\min_{f \in \mathcal{F}} R^*(f) = \mathbb{E}_{p_X} \mu^*(X)$. Then for any $\delta \in (0, 1/2)$, with probability at least $1 - \delta$,*

$$R^*(\hat{f}_T) - \mathbb{E}_{p_X} \mu^*(X) \lesssim M \left( \left( \frac{\gamma_T^{avg}}{T} \right)^{\frac{1+\nu}{2+\nu(1+p/2)}} + \left( \frac{\gamma_T^{\max}}{T} \right)^{\frac{1+\nu}{1+\nu(1+p/2)}} \right.$$

$$\left. + \left( \frac{\gamma_T^{avg} \log(1/\delta)}{T} \right)^{\frac{1+\nu}{2+\nu}} + \frac{\gamma_T^{\max} \log(1/\delta)}{T} \right).$$

**Remark 5** (Classification using ISWERM). The above results can easily be rephrased for the classification analogue to the regression problem in Section 4, where $\mathcal{Y} = \{\pm 1\}$ and we want a classifier based on features $x, a$ to minimize misclassification error. Because the policy learning problem is both of greater interest and greater generality, we focus our presentation on policy learning.

## 6 Empirical Study

Next, we empirically investigate various risk minimization approaches using data collected by a contextual bandit algorithm, including both ISWERM and unweighted ERM among others. We take 51 different mutli-calss classification datasets from OpenML-CC18 [11] and transform each into a multi-arm contextual bandit problem (following the approach of [15, 17, 51]). We then run an epsilon greedy algorithm for $T = 100000$, where we explore uniformly with probability $\epsilon_t = t^{-1/3}$ and otherwise pull the arm that maximizes an estimate of $\mu(x, a)$ based on data so far. Details are given in Appendix F.1.

We then consider using this data to regress $Y_t$ on $X_t, A_t$ using different methods where each observation is weighted by $w_t$ using different schemes: (1) Unweighted ERM: $w_t = 1$; (2) ISWERM:

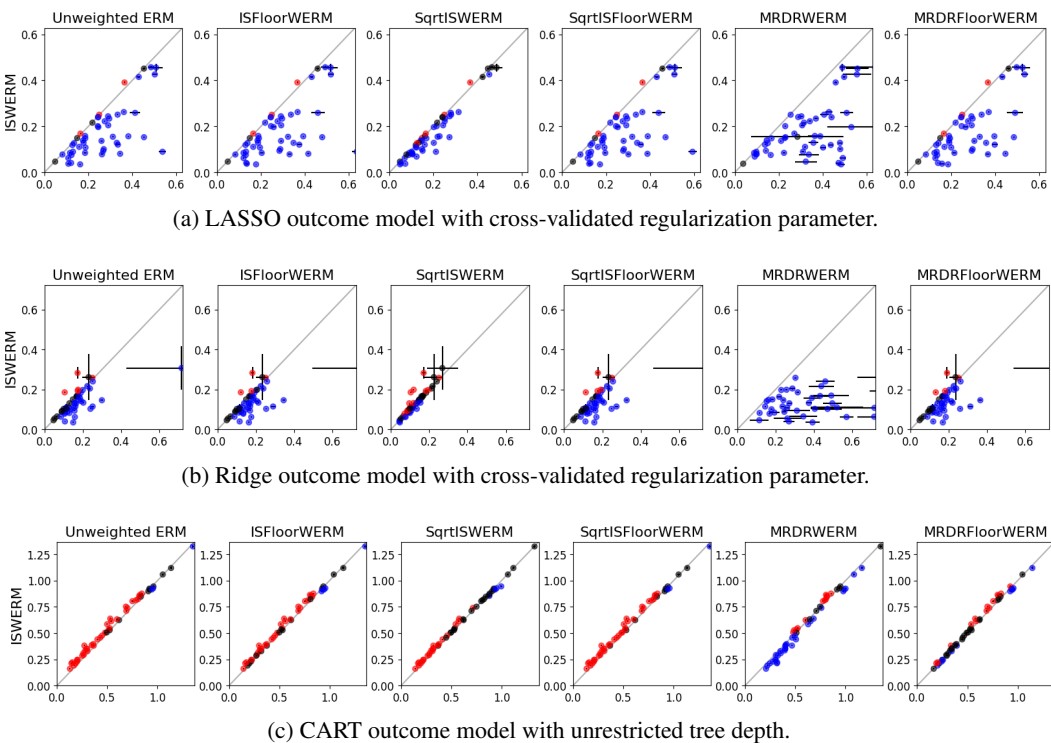

(a) LASSO outcome model with cross-validated regularization parameter.

(b) Ridge outcome model with cross-validated regularization parameter.

(c) CART outcome model with unrestricted tree depth.

Figure 1: Comparison of weighted regression run on contextual-bandit-collected data. Each dot is one of 51 OpenML-CC18 datasets. Lines denote $\pm 1$ standard error. Dots are blue when ISWERM is clearly better, red when clearly worse, and black when indistinguishable within one standard error.

$w_t = g_t^{-1}(A_t \mid X_t)$; (3) ISFloorWERM: $w_t = \gamma_t^{-1}$, where–inspired by [58]–we weight by the inverse (nonrandom) floor $\gamma_t = \epsilon_t/|\mathcal{A}|$ of the propensity scores; (4) SqrtISWERM: $w_t = g_t^{-1/2}(A_t \mid X_t)$, which applies the stabilization of [24, 38] to ISWERM; (5) SqrtISFloorWERM: $w_t = \gamma_t^{-1/2}$; (6) MRDRWERM: $w(t) = \frac{1-g_t(A_t|X_t)}{g_t^2(A_t|X_t)}$, which are the weights used by Farajtabar et al. [18]; (7) MRDRFloorWERM: $w(t) = \frac{1-\gamma_t}{\gamma_t^2}$, which is like MRDRWERM but uses the propensity score floors $\gamma_t$. With these sample weights, we run either Ridge regression, LASSO, or CART using sklearn's `RidgeCV(cv=4)`, `LassoCV(cv=4)`, or `DecisionTreeRegressor`, each with default parameters. For Ridge and LASSO we pass as features the intercept-augmented contexts $\{(1, X_t)\}_{t=1}^T$ concatenated by the product of the one-hot encoding of arms $\{A_t\}_{t=1}^T$ with the intercept-augmented contexts $\{(1, X_t)\}_{t=1}^T$. For CART, we use the concatenation of the contexts $\{X_t\}_{t=1}^T$ with the one-hot encoding of arms $\{A_t\}_{t=1}^T$. To evaluate, we play our bandit anew for $T^{\text{test}} = 1000$ rounds using a uniform exploration policy, $g^*(a \mid x) = 1/K$, and record the mean-squared error (MSE) of the regression fits on this data. We repeat the whole process 64 times and report estimated average MSE and standard error in Fig. 1.

**Results.** Figures 1a and 1b show that ISWERM clearly outperforms unweighted ERM and all other weighted-ERM schemes for linear regression, with ISWERM's advantage being even more pronounced for LASSO. Intuitively, since a linear model is misspecified, this can be attributed to ISWERM's ability to provide agnostic best-in-class risk guarantees. In contrast, for a better specified model such as CART, all ERM methods perform similarly, as seen in Fig. 1c. We highlight that our focus is not necessarily methodological improvements, and the aim of our experiments is to explore the implications of our theory, not provide state-of-the-art results. We provide additional empirical results in Appendix F.2, the conclusions from which are qualitatively the same.

# 7 Conclusions and Future Work

We provided first-of-their-kind guarantees for risk minimization from adaptively collected data using ISWERM. Most crucially, our guarantees provided good dependence on the size of IS weights leading to correct convergence rates when exploration diminishes with time, as happens when we collect data using a contextual bandit algorithm. This was made possible by a new maximal inequality specifically for IS weighted sequential empirical processes. There are several important avenues for future work. We focused on a fixed hypothesis class. One important next question is how to do effective model selection in adaptive settings. We also focused on IS weighted regression and policy learning, but recent work in the iid setting highlights the benefits of using doubly-robust-style centering [2, 21, 33]. These benefits are most important to avoid rate deterioration when IS weights are estimated, while our IS weights are known, but there are still benefits in reducing the loss variance in the leading constant. Therefore, exploring such methods in adaptive settings is another important next question.

# 8 Societal Impact

Our work provides guarantees for learning from adaptively collected data. While the methods (IS weighting) are standard, our novel guarantees lend credibility to the use of adaptive experiments. Adaptive experiments hold great promise for better, more efficient, and even more ethical experiments. At the same time, adaptive experiments, especially when all arms are always being explored ($\gamma_t < \infty$) even if at vanishing rates ($\gamma_t = \omega(1)$), must still be subject to the same ethical guidelines as classic randomized experiments regarding favorable risk-benefit ratio of any arm, informed consent, and other protections of participants. There are also several potential dangers to be aware of in supervised and policy learning generally, such as the training data possibly being unrepresentative of the population to which predictions and policies will be applied leading to potential disparities as well as the focus on *average* welfare compared to prediction error or policy value on each individual or group. These remain concerns in the adaptive setting, and while ways to tackle these challenges in non-adaptive settings might be applicable in adaptive ones, a rigorous study of applicability requires future work.

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
