Supplementary Material for:
# Risk Minimization from Adaptively Collected Data:
## Guarantees for Supervised and Policy Learning

## A  Proof of the maximal inequality for IS weighted sequential empirical processes

### A.1  Preliminary lemmas

For any sequence $\widetilde{g}_1, \ldots, \widetilde{g}_T$ of conditional densities and any finite sequence $\zeta_{1:T} := (\zeta_t)_{t=1}^T$ of $\mathcal{O} \to \mathbb{R}$ functions, let

$$\rho_{T,\widetilde{g}_{1:T}}(\zeta_{1:T}) := \left( \frac{1}{T} \sum_{t=1}^T \|\zeta_t\|_{2,\widetilde{g}_t}^2 \right)^{1/2}.$$

For any conditional density $\widetilde{g} : (a, x) \in \mathcal{A} \times \mathcal{X} \mapsto \widetilde{g}(a \mid x)$, let

$$\rho_{T,\widetilde{g}}(\zeta_{1:T}) := \rho_{T,\widetilde{g}_{1:T}}(\zeta_{1:T}),$$

where we set $\widetilde{g}_t := \widetilde{g}$ for every $t \in [T]$.

**Lemma 4.** *Any $\rho_{T,\widetilde{g}_{1:T}}$ as defined above is a pseudonorm over the vector space $(\mathcal{O} \to \mathbb{R})^T$.*

*Proof of Lemma 4.* It is immediate that for any real number $\lambda$, and finite sequence $\zeta_{1:T}$ of $\mathcal{O} \to \mathbb{R}$ functions $\rho_{T,\widetilde{g}_{1:T}}(\lambda \zeta_{1:T}) = |\lambda| \rho_{T,\widetilde{g}_{1:T}}(\zeta_{1:T})$.

We now check that $\rho_{T,\widetilde{g}_{1:T}}$ satisfies the triangle inequality. Let $\zeta_{1:T}^{(1)}$ and $\zeta_{1:T}^{(2)}$ be two sequences of $\mathcal{O} \to \mathbb{R}$ functions. We have that

$$\begin{aligned}
\rho_{T,\widetilde{g}_{1:T}}(\zeta_{1:T}^{(1)} + \zeta_{1:T}^{(2)}) &= \left( \frac{1}{T} \sum_{t=1}^T \left\| \zeta_t^{(1)} + \zeta_t^{(2)} \right\|_{2,\widetilde{g}_t}^2 \right)^{1/2} \\
&\leq \left( \frac{1}{T} \sum_{t=1}^T \left( \left\| \zeta_t^{(1)} \right\|_{2,\widetilde{g}_t} + \left\| \zeta_t^{(2)} \right\|_{2,\widetilde{g}_t} \right)^2 \right)^{1/2} \\
&\leq \left( \frac{1}{T} \sum_{t=1}^T \left\| \zeta_t^{(1)} \right\|_{2,\widetilde{g}_t}^2 \right)^{1/2} + \left( \frac{1}{T} \sum_{t=1}^T \left\| \zeta_t^{(2)} \right\|_{2,\widetilde{g}_t} \right)^{1/2} \\
&= \rho_{T,\widetilde{g}_{1:T}}(\zeta_{1:T}^{(1)}) + \rho_{T,\widetilde{g}_{1:T}}(\zeta_{1:T}^{(2)}),
\end{aligned}$$

where the second line above follows from the triangle inequality applied to the pseudonorms $\| \cdot \|_{2,\widetilde{g}_t}$, $t = 1, \ldots, T$, and where the third line follows from the triangle inequality applied to the Euclidean norm $x \in \mathbb{R}^T \mapsto (\sum_{t=1}^T x_t^2)^{1/2}$. $\qquad\square$

**Lemma 5.** *Consider $g^*$ and $g_1, \ldots, g_T$ as defined in the main text. Suppose that assumption 1 holds. Then, for any finite sequence of functions $(\zeta_t)_{t=1}^T \in (\mathcal{O} \to \mathbb{R})^T$,*

$$\rho_{T,g_{1:T}}\left( \frac{g^*}{g_{1:T}} \zeta_{1:T} \right) \leq \sqrt{\gamma_T^{\max}} \rho_{T,g^*}(\zeta_{1:T}).$$

*If all elements of the sequence $\zeta_t$ are the same, that is, if there exists $\zeta : \mathcal{O} \to \mathbb{R}$ such that $\zeta_t = \zeta$ for every $t \in [T]$, then*

$$\rho_{T,g_{1:T}}\left( \frac{g^*}{g_{1:T}} \zeta_{1:T} \right) \leq \sqrt{\gamma_T^{avg}} \|\zeta\|_{2,g^*}.$$

*Proof of Lemma 5.* We have that

$$\rho_{T,g_{1:T}}\left(\frac{g^*}{g_{1:T}}\zeta_{1:T}\right) = \left(\frac{1}{T}\sum_{t=1}^{T}P_{g_t}\left(\frac{g^*}{g_t}\zeta_t\right)^2\right)^{1/2}$$

$$= \left(\frac{1}{T}\sum_{t=1}^{T}P_{g^*}\left(\frac{g^*}{g_t}\zeta_t^2\right)\right)^{1/2}$$

$$\leq \left(\frac{1}{T}\sum_{t=1}^{T}\gamma_t P_{g^*}\zeta_t^2\right)^{1/2},$$

where the inequality follows from Assumption 1. If there exists $\zeta : \mathcal{O} \to \mathbb{R}$ such that $\zeta_t = \zeta$ for every $t = 1, \ldots, T$, then,

$$\rho_{T,g_{1:T}} \leq \left(\frac{1}{T}\sum_{t=1}^{T}\gamma_t P_{g^*}\zeta_t^2\right)^{1/2}$$

$$= \sqrt{\gamma_T^{\mathrm{avg}}}\,\|\zeta\|_{2,g^*}.$$

Otherwise, we have

$$\rho_{T,g_{1:T}} \leq \left(\frac{1}{T}\sum_{t=1}^{T}\gamma_t P_{g^*}\zeta_t^2\right)^{1/2}$$

$$\leq \left(\max_{t\in[T]}\gamma_t\frac{1}{T}\sum_{t=1}^{T}P_{g^*}\zeta_t^2\right)^{1/2}$$

$$= \sqrt{\gamma_T^{\mathrm{max}}}\,\rho_{T,\tilde{g}_{1:T}}(\zeta_{1:T}).$$

$\square$

The following lemma is a restatement under our notation of Corollary A.8 in van Handel [57].

**Lemma 6.** *Let $\zeta_{1:T}^1, \ldots, \zeta_{1:T}^N$ be $N$ $\bar{O}_{1:T}$-predictable sequences of $\mathcal{O} \to \mathbb{R}$ functions, and let $A$ be an $\bar{O}_T$-measurable event. Then, for any $r > 0$ and any $b > 0$ such that $\max_{i\in[N],t\in[T]}\|\zeta_t^i\|_\infty \leq b$, it holds that*

$$E\left[\max_{i\in[N]}\frac{1}{T}\sum_{t=1}^{N}(\delta_{O_t} - P_{g_t})\zeta_t^i\mathbf{1}(\rho_{T,g_{1:T}}(\zeta_{1:T}^i) \leq r) \mid A\right]$$

$$\lesssim r\sqrt{\frac{\log(1 + N/P[A])}{t}} + \frac{B}{t}\log(1 + N/P[A]).$$

### A.2 Proof of Theorem 1

*Proof of Theorem 1.* We treat together both the general case where, for each $f$, $\xi_{1:T}(f)$ is an $\bar{O}_{1:T}$-predictable sequence, and the case where, for every $f$, there exists a deterministic $\xi(f) : \mathcal{O} \to \mathbb{R}$ such that $\xi_t(f) = \xi(f)$ for every $t \in [T]$. We refer to the former as *case 1* and to the latter as *case 2* in the rest of the proof. In case 1, we let $\tilde{\rho}_T := \rho_T^{\mathrm{max}}$, and in case 2, we let $\tilde{\rho}_T := \bar{\rho}_T$.

**From a conditional expectation bound to a high probability bound.** Let $x > 0$. We introduce the following event:

$$A := \left\{\sup_{f\in\mathcal{F}} M_T(f) \geq \psi(x)\right\},$$

where

$$\psi(x) := C\left\{ r^- + \sqrt{\frac{\widetilde{\gamma_T}}{T}} \int_{r^-}^r \sqrt{\log(1 + \mathcal{N}_{[\,]}(\epsilon, \Xi_T, \rho_{T,g^*}))}d\epsilon \right.$$

$$+ \frac{B\gamma_T^{\max}}{T} \log(1 + \mathcal{N}_{[\,]}(r, \Xi_T, \rho_{T,g^*}))$$

$$\left. r\sqrt{\frac{x}{T}} + \frac{\gamma_T^{\max} x}{T} \right\},$$

where $C$ is a universal constant to be discussed further down. Suppose we can show that

$$E\left[\sup_{f \in \mathcal{F}} M_T(f) \mid A\right] \leq \psi\left(\log\left(1 + \frac{1}{P[A]}\right)\right).$$

Then, we will have that $\psi(x) \leq \psi(\log(2/P[A]))$, that is $P[A] \leq 2e^{-x}$, which is the wished claim.

**Setting up the chaining decomposition.** Let $\epsilon_0 := r$, and, for every $j \geq 0$, let $\epsilon_j := \epsilon_0 2^{-j}$. For any $j \geq 0$, let

$$\mathcal{B}^j := \left\{ (\lambda_s^{j,k}, \upsilon_t^{j,k})_{t=1}^T : k \in [N_j] \right\}$$

be a minimal $(\epsilon_j, \rho_{T,g^*})$-sequential bracketing of $\Xi_T$. For any $f \in \mathcal{F}$, let $k(j, f) \in [N_j]$ be such that

$$\lambda_s^{j,k(j,f)} \leq \xi_t(f) \leq \upsilon^{j,k(j,f)} \text{ for every } t \in [T],$$

and let $\Delta_t^{j,f} := \upsilon_s^{j,k(j,f)} - \lambda_s^{j,k(j,f)}$ and $u^{j,f} := \upsilon_s^{j,k(j,f)}$. For any $j \geq 0$, let $\bar{N}_j := \prod_{i=0}^j N_i$. For any $j \geq 0$, and $t \in [T]$ let

$$a_{j,t} := \epsilon_j \sqrt{\frac{T}{\log(1 + \bar{N}_j/P[A])}} \frac{\sqrt{\widetilde{\gamma_T}}}{\gamma_t}.$$

Let $J \geq 0$ such that $\epsilon_{J+1} < r^- \leq \epsilon_J$. The integer $J$ will be the maximal depth of the chains in our chaining decomposition. For any $t \in [T]$, $f \in \mathcal{F}$, let

$$\tau_t(f) := \inf\left\{ j \geq 0 : \Delta_t^{j,f} > a_{j,t} \right\} \wedge J,$$

be the depth at which we truncate the chains, adaptively depending on the value of $\Delta_t^{j,f}$, so that $\Delta_t^{j,f}\mathbf{1}(\tau_t(f) > j)$ is no larger than $a_{j,t}$ in supremum norm at any depth $j$.

For any $f \in \mathcal{F}$ and any $t \in [T]$, the following chaining decomposition holds:

$$\xi_t(f) = \underbrace{\sum_{j=0}^J (\xi_t(f) - u^{j,f} \wedge u^{j-1,f})\mathbf{1}(\tau_t(f) = j)}_{\text{tip of the chain}}$$

$$+ \underbrace{\sum_{j=1}^J \left\{ (u^{j,f} \wedge u^{j-1,f} - u^{j-1,f})\mathbf{1}(\tau_t(f) = j) + (u^{j,f} - u^{j-1,f})\mathbf{1}(\tau_t(f) > j) \right\}}_{\text{links of the chain}}$$

$$+ \underbrace{u_t^{0,f}}_{\text{root of then chain}}.$$

**Control of the tips.**

- **Case $j = J$.** We have that

$$\frac{1}{T}\sum_{t=1}^{T}(\delta_{O_t} - P_{g_t})\frac{g^*}{g_t}(\xi_t(f) - u_t^{J,f} \wedge u_t^{J-1,f})\mathbf{1}(\tau_t(f) = J)$$

$$\leq \frac{1}{T}\sum_{t=1}^{T}P_{g_t}\frac{g^*}{g_t}\Delta_t^{J,f}$$

$$= \frac{1}{T}\sum_{t=1}^{T}\|\Delta_t^{J,f}\|_{1,g^*}$$

$$\leq \left(\frac{1}{T}\sum_{t=1}^{T}\|\Delta_t^{J,f}\|_{2,g^*}^2\right)^{1/2}$$

$$\leq \epsilon_J.$$

Therefore

$$E\left[\sup_{f\in\mathcal{F}}\frac{1}{T}\sum_{t=1}^{T}(\delta_{O_t-P_{g_t}}\frac{g^*}{g_t})(\xi_t(f) - u_t^{J,f} \wedge u_t^{J-1,f})\mathbf{1}(\tau_t(f) = J) \mid A\right] \leq \epsilon_J.$$

- **Case $j < J$.**

$$\frac{1}{T}\sum_{t=1}^{T}(\delta_{O_t} - P_{g_t})\frac{g^*}{g_t}(\xi_t(f) - u_t^{j,f} \wedge u_t^{j-1,f})\mathbf{1}(\tau_t(f) = j)$$

$$\leq \frac{1}{T}\sum_{t=1}^{T}P_{g_t}\frac{g^*}{g_t}\Delta_t^{j,f}\mathbf{1}(\tau_t(f) = j)$$

$$\leq \frac{1}{T}\sum_{t=1}^{T}P_{g^*}\frac{(\Delta_t^{j,f})^2}{a_{j,t}}$$

$$\leq \epsilon_j^2\frac{1}{T}\sum_{t=1}^{T}\frac{1}{a_{j,t}}$$

$$= \epsilon_j\sqrt{\frac{\log(1 + \bar{N}_j/P[A])}{T}}\frac{1}{\sqrt{\widetilde{\gamma}_T}}\frac{1}{T}\sum_{t=1}^{T}\gamma_t$$

$$\leq \epsilon_j\sqrt{\frac{\widetilde{\gamma}_T\log(1 + \bar{N}_j/P[A])}{T}}.$$

(The last inequality is an equality in *case 2*).

**Control of the links.** We start with bounding the $\rho_{T,g_{1:T}}$ pseudo-norm of the IS weighted links. We have that

$$\rho_{T,g_{1:T}}\left(\left(\frac{g^*}{g_t}(u_t^{j,f} \wedge u_t^{j-1,f} - u_t^{j-1,f})\right)_{t=1}^{T}\right)$$

$$\leq \rho_{T,g_{1:T}}\left(\left(\frac{g^*}{g_t}(u_t^{j,f} - u_t^{j-1,f})\right)_{t=1}^{T}\right)$$

$$\leq \sqrt{\widetilde{\gamma}_T}\rho_{T,g^*}\left(u_{1:T}^{j,f} - u_{1:T}^{j-1,f}\right)$$

$$\leq \sqrt{\widetilde{\gamma}_T}\left\{\rho_{T,g^*}\left(u_{1:T}^{j,f} - \xi_{1:T}(f)\right) + \rho_{T,g^*}\left(\xi_{1:T}(f) - u_{1:T}^{j-1,f}\right)\right\}$$

$$\lesssim \sqrt{\widetilde{\gamma}_T}\epsilon_j,$$

where we have used lemma 5 is the third line and where the fourth line above follows from the triangle inequality.

We now bound the supremum norm of the links. For every $t \in [T]$,

$$(u_t^{j,f} \wedge u_t^{j-1,f} - u_t^{j,f})\mathbf{1}(\tau_t(f) = j)$$
$$= (u_t^{j,f} \wedge u_t^{j-1,f} - \xi_t(f))\mathbf{1}(\tau_t(f) = j)$$
$$- (u_t^{j-1,f} - \xi(f))\mathbf{1}(\tau_t(f) = j).$$

Using the definition of $\tau_t(f)$, we obtain

$$0 \leq (u_t^{j,f} \wedge u_t^{j-1,f} - \xi_t(f))\mathbf{1}(\tau_t(f) = j) \leq (u_t^{j-1,f} - \xi_t(f))\mathbf{1}(\tau_t(f) = j) \leq a_{j-1,t} \lesssim a_{j,t},$$

and

$$0 \leq (u_t^{j-1,f} - \xi_t(f))\mathbf{1}(\tau_t(f) = j) \leq a_{j-1,t} \lesssim a_{j-1,t}.$$

Therefore,

$$\max_{t \in [T]} \left\| \frac{g^*}{g_t} \left( u_t^{j,f} \wedge u_t^{j-1,f} - u_t^{j-1,f} \right) \mathbf{1}(\tau_t(f) = j) \right\|_\infty \lesssim \gamma_t a_{j,t} = b_j$$

where

$$b_j := \epsilon_j \sqrt{\frac{T\widetilde{\gamma}_T}{\log(1 + \bar{N}_j/P[A])}}$$

Similarly, we have

$$0 \leq (u_t^{j,f} - \xi_t(f))\mathbf{1}(\tau_t(f) > j) \leq a_{j,t} \qquad \text{and} \qquad 0 \leq (u_t^{j-1,f} - \xi_t(f))\mathbf{1}(\tau_t(f) > j) \leq a_{j-1,t},$$

and therefore, for every $t \in [T]$

$$\left\| \frac{g^*}{g_t} \left( u_t^{j-1,f} - u_t^{j-1,f} \right) \mathbf{1}(\tau_t(f) > j) \right\|_\infty \lesssim \gamma_t a_{j,t} = b_j$$

Denote

$$v_t^{j,f} := \frac{g^*}{g_t} \left\{ (u_t^{j,f} \wedge u_t^{j-1,f} - u_t^{j,f})\mathbf{1}(\tau_t(f) = j) + (u_t^{j,f} - u_t^{j-1,f})\mathbf{1}(\tau_t(f) > j) \right\}.$$

Observe that as $f$ varies over $\mathcal{F}$, $v_{1:T}^{j,f}$ varies over a collection of at most $N_j \times N_{j-1} \leq \bar{N}_j$ elements. Therefore, lemma 6 yields

$$E \left[ \sup_{f \in \mathcal{F}} \frac{1}{T} \sum_{t=1}^{T} (\delta_{O_t} - P_{g_t}) \frac{g^*}{g_t} v_t^{j,f} \right]$$
$$\lesssim \epsilon_j \sqrt{\frac{\widetilde{\gamma}_T \log(1 + \bar{N}_j/P[A])}{T}} + \frac{b_j}{T} \log(1 + \bar{N}_j/P[A])$$
$$\lesssim \epsilon_j \sqrt{\frac{\widetilde{\gamma}_T \log(1 + \bar{N}_j/P[A])}{T}}.$$

**Control of the root.** For any $f$ such that $\rho_{T,g^*}((\xi_t(f))_{t=1}^T) \leq r$, we have that

$$\rho_{T,g_{1:T}}(((g^*/g_t)u_t^{0,f})_{t=1}^T)$$
$$\leq \sqrt{\widetilde{\gamma}_T} \rho_{T,g^*}(u_{1:T}^{0,f})$$
$$\leq \sqrt{\widetilde{\gamma}_T}(\rho_{T,g^*}(u_{1:T}^{0,f} - \xi_{1:T}(f)) + \rho_{T,g^*}(\xi_{1:T}(f))).$$

Without loss of generality, we can assume that $\max_{t \in [T]} \|u_t^{0,f}\|_\infty \leq B$, since thresholding to $B$ preserves the bracketing property. Therefore, $\max_{t \in [T]} \|(g^*/g_t)u_t^{0,f}\|_\infty \leq \gamma_T^{\max} B\epsilon$.

Then, from lemma 6,

$$E \left[ \sup \left\{ \frac{1}{T} \sum_{t=1}^{T} (\delta_{O_t} - P_{g_t})\xi_t(f) : f \in \mathcal{F}, \rho_{T,g^*}((\xi_t(f))_{t=1}^T) \leq r \right\} \right]$$
$$\leq \sqrt{\frac{\widetilde{\gamma}_T}{T}} \sqrt{\log \left( (1 + \frac{\bar{N}_0}{P[A]}) \right)} + \frac{B\gamma_T^{\max}}{T} \log \left( 1 + \frac{\bar{N}_0}{P[A]} \right)$$

**Adding up the bounds.** We obtain

$$
E\left[\sup_{f\in\mathcal{F}} M_T(f)\mid A\right] \lesssim \underbrace{\sqrt{\frac{\widetilde{\gamma_T}}{T}}\sqrt{\log\left(\left(1+\frac{\bar{N}_0}{P[A]}\right)\right)}+\frac{B}{\delta T}\log\left(1+\frac{\bar{N}_0}{P[A]}\right)}_{\text{root contribution}}
$$

$$
+\underbrace{\sqrt{\frac{\widetilde{\gamma_T}}{T}}\sum_{j=1}^{J}\epsilon_j\log\left(1+\frac{\bar{N}_j}{P[A]}\right)}_{\text{links contribution}}
$$

$$
+\underbrace{\sqrt{\frac{\widetilde{\gamma_T}}{T}}\sum_{j=0}^{J-1}\epsilon_j\log\left(1+\frac{\bar{N}_j}{P[A]}\right)+\epsilon_J}_{\text{tip contribution}}
$$

$$
\lesssim \epsilon_J+\sqrt{\frac{\widetilde{\gamma_T}}{T}}\sum_{j=0}^{J}\epsilon_j\log\left(1+\frac{\bar{N}_j}{P[A]}\right)+\frac{B\gamma_T^{\max}}{T}\log\left(1+\frac{\bar{N}_0}{P[A]}\right).
$$

We use the classical technique from finite adaptive chaining proofs to bound the sum in the second term with an integral [see e.g. 8, 57]. We obtain

$$
\sum_{j=0}^{J}\epsilon_j\log\left(1+\frac{\bar{N}_j}{P[A]}\right)\lesssim\int_{r^-}^{r}\sqrt{\log(1+N_{[]}(\epsilon,\Xi_T,\rho_{T,g^*}))}d\epsilon+\log\left(1+\frac{1}{P[A]}\right).
$$

Therefore,

$$
E\left[\sup_{f\in\mathcal{F}} M_T(f)\mid A\right] \lesssim r^-+\sqrt{\frac{\widetilde{\gamma_T}}{T}}\int_{r^-}^{r}\sqrt{\log(1+N_{[]}(\epsilon,\Xi_T,\rho_{T,g^*}))}d\epsilon
$$

$$
+\frac{B\gamma_T^{\max}}{T}\log(1+N_{[]}(r,\Xi_T,\rho_{T,g^*}))
$$

$$
+\sqrt{\frac{\widetilde{\gamma_T}}{T}}\sqrt{\log\left(1+\frac{1}{P[A]}\right)}+\frac{B\gamma_T^{\max}}{T}\log\left(1+\frac{1}{P[A]}\right).
$$

Therefore, for an appropriate choice of the universal constant $C$ in the definition of $\psi$, we have that

$$
E\left[\sup_{f\in\mathcal{F}} M_T(f)\mid A\right]\leq\psi\left(\log\left(1+\frac{1}{P[A]}\right)\right),
$$

which, from the first paragraph of the proof, implies the wished claim. $\qquad\square$

# B  Proof of the excess risk bounds for ISWERM

## B.1  Proof of Theorem 2

*Proof of Theorem 2.* Let

$$
M_T(f):=\frac{1}{T}\sum_{t=1}^{T}(P_{g_t}-\delta_{O_t})(\ell(f)-\ell(f_1)).
$$

Since $\widehat{R}_T(\widehat{f}_T)-\widehat{R}_T(f_1)\leq 0$, and from the diameter assumption 3, we have that $R^*(\widehat{f}_T)-R^*(f_1)\leq \sup\{M_T(f):f\in\mathcal{F},\|\ell(f)-\ell(f_1)\|_{2,g^*}\leq\rho_0\|\Lambda\|_{2,g^*}\}$. Therefore, from the diameter assumption (Assumption 3), Theorem 1 yields, via the change of variable $r=\rho\|\Lambda\|_{2,g^*}$, for any $x>0$,

$\rho^- \in [0, \rho_0/2]$, that it holds with probability at least $1 - 2e^{-x}$ that

$$R^*(\widehat{f}_T) - R^*(f_1) \leq \|\Lambda\|_{2,g^*} \left\{ \rho^- + \sqrt{\frac{\gamma_T^{\mathrm{avg}}}{T}} \int_{\rho^-}^{\rho_0} \sqrt{\log(1 + N_{[]}(\epsilon\|\Lambda\|_{2,g^*}, \ell(\mathcal{F}), \|\cdot\|_{2,g^*})} d\epsilon \right.$$
$$+ \frac{b_0 \gamma_T^{\max}}{T} \log(1 + N_{[]}(\rho_0\|\Lambda\|_{2,g^*}, \ell(\mathcal{F}), \|\cdot\|_{2,g^*}))$$
$$\left. + \sqrt{\frac{\gamma_T^{\mathrm{avg}} x}{T}} + \frac{b_0 \gamma_T^{\max} x}{T} \right\}.$$

In the case $p \in (0, 2)$, setting $\rho^- = 0$ and $x = \log(1/\delta)$, and plugging in the entropy assumption (Assumption 2) immediately yield the claim. In the case $p > 2$, setting $x = \log(1/\delta)$, plugging in the entropy assumption and optimizing the value of $\rho^-$ yields the claim. $\qquad\square$

## B.2 Proof of Theorem 3

*Proof.* From convexity of $f \mapsto \ell(f, \cdot)$ and of $\mathcal{F}$, the following implication holds, for any $r > 0$:

$$\exists f \in \mathcal{F}, \ R^*(f) - R^*(f_1) \geq r^2 \qquad \text{and} \widehat{R}_T(f) - \widehat{R}_T(f_1) \leq 0$$
$$\implies \exists f \in \mathcal{F}, \ R^*(f) - R^*(f_1) = r^2 \qquad \text{and} \widehat{R}_T(f) - \widehat{R}_T(f_1) \leq 0.$$

Let

$$M_T(f) := \frac{1}{T} \sum_{t=1}^T (P_{g_t} - \delta_{O_t})(\ell(f) - \ell(f_1)).$$

Let $\rho > 0$. Since $\widehat{R}_T(\widehat{f}_T) - \widehat{R}_T(f_1) \leq 0$, we have that

$$P\left[ R^*(\widehat{f}_T) - R^*(f_1) \geq \rho^2 \|\Lambda\|_{2,g^*} \|_{2,g^*} \right]$$
$$\leq P\left[ \exists f \in \mathcal{F}, R^*(f) - R^*(f_1) = \rho^2 \|\Lambda\|_{2,g^*} \text{ and } \widehat{R}_T(f) - \widehat{R}_T(f_1) \leq 0 \right]$$
$$\leq P\left[ \exists f \in \mathcal{F}, \sup\{M_T(f) : f \in \mathcal{F}, \|\ell(f) - \ell(f_1)\|_{2,g^*}\} \lesssim \|\Lambda\|_{2,g^*} \rho^\alpha \right],$$

where we have used the variance bound in the last line (Assumption 4). From theorem 1 and the loss diameters assumption 3, we have, for any $x, \rho > 0$, that it holds with probability at least $1 - 2e^{-x}$ that

$$\sup\{M_T(f) : f \in \mathcal{F}, \|\ell(f) - \ell(f_1)\|_{2,g^*}\} \lesssim \psi_T(\rho),$$

with

$$\psi_T(\rho) := \|\Lambda\|_{2,g^*} \left\{ \sqrt{\frac{\gamma_T^{\mathrm{avg}}}{T}} \int_0^{\rho^\alpha} \sqrt{\log(1 + N_{[]}(\epsilon\|\Lambda\|_{2,g^*}, \ell(\mathcal{F}), \|\cdot\|_{2,g^*}))} d\epsilon \right.$$
$$+ \frac{b_0 \gamma_T^{\max}}{T} \log(1 + N_{[]}(\rho^\alpha \|\Lambda\|_{2,g^*}, \ell(\mathcal{F}), \|\cdot\|_{2,g^*})$$
$$\left. + \rho^\alpha \sqrt{\frac{\gamma_T^{\mathrm{avg}} x}{T}} + \frac{b_0 \gamma_T^{\max} x}{T} \right\}.$$

Therefore, if $\rho$ is such that $\rho^2 \|\Lambda\|_{2,g^*} \geq \psi_T(\rho)$, then with probability at least $1 - 2e^{-x}$,

$$R^*(\widehat{f}_T) - R^*(f_1) \lesssim \rho^2 \|\Lambda\|_{2,g^*}.$$

We therefore compute an upper bound on $\psi_T(\rho)$. From the entropy assumption (2), we have that

$$\psi_T(\rho) \lesssim \|\Lambda\|_{2,g^*} \left\{ \sqrt{\frac{\gamma_T^{\mathrm{avg}}}{T}} \rho^{\alpha(1-p/2)} + \frac{b_0 \gamma_T^{\max}}{T} \rho^{-p\alpha} + \rho^\alpha \sqrt{\frac{\gamma_T^{\mathrm{avg}} x}{T}} + \frac{b_0 \gamma_T^{\max}}{T} x \right\}.$$

Therefore, a sufficient condition for $\rho^2 \|\Lambda\|_{2,g^*} \geq \psi_T(\rho)$ is that

$$\rho^2 \geq \max\left\{ \sqrt{\frac{\gamma_T^{\mathrm{avg}}}{T}} \rho^{\alpha(1-p/2)}, \frac{b_0 \gamma_T^{\max}}{T} \rho^{-p\alpha}, \rho^\alpha \sqrt{\frac{\gamma_T^{\mathrm{avg}} x}{T}}, \frac{b_0 \gamma_T^{\max}}{T} x \right\}$$

that is

$$\rho^2 \geq \max \left\{ \left( \frac{\gamma_T^{\mathrm{avg}}}{T} \right)^{\frac{1}{2-\alpha+p\alpha/2}}, \left( \frac{\gamma_T^{\mathrm{avg}}}{T} \right)^{\frac{1}{2-\alpha}}, \left( \frac{b_0 \gamma_T^{\mathrm{max}}}{T} \right)^{\frac{1}{1+p\alpha/2}}, \frac{b_0 \gamma_T^{\mathrm{max}} x}{T} \right\},$$

which immediately implies the wished claim. $\qquad\square$

## C   Proof of the results on least squares regression using ISWERM

*Proof of lemma 2.* For any $o = (x, a, y) \in \mathcal{O}$, $f, f' : \mathcal{O} \to \mathbb{R}$, we have

$$|\ell(f)(o) - \ell(f')(o)| = |2y - f(a, x) - f'(a, x)||f(a, x) - f_1(a, x)|$$
$$\leq 4\sqrt{M}|f(a, x) - f_1(a, x)|,$$

which is the second claim. This inequality further gives that, for any $f \in \mathcal{F}$

$$\|\ell(f) - \ell(f_1)\|_{2, g^*} \leq 4\sqrt{M} \|f - f_1\|_{2, g^*}.$$

We now show that $\|f - f_1\|_{2, g^*} \leq R^*(f) - R^*(f_1)$. Recall the definition of $\mu$: for any $(a, x) \in \mathcal{A}, \mathcal{X}$, $\mu(a, x) := E_{p_Y}[Y \mid A = a, X = x]$. From Pythagoras, $R^*(f) = E[(Y - \mu(A, X))^2] + \|\mu - f\|_{2, g^*}^2$. For any $h_1, h_2 : \mathcal{A} \times \mathcal{X} \to \mathbb{R}$, denote $\langle h_1, h_2 \rangle := E_{p_X, g^*}[h_1(A, X) h_2(A, X)]$. We have that

$$R^*(f) - R^*(f_1) - \|f - f_1\|_{2, g^*}$$
$$= \|f - \mu\|_{2, g^*}^2 - \|f_1 - \mu\|_{2, g^*}^2 - \|f - f_1\|_{2, g^*}^2$$
$$= \langle f - f_1, f_1 - \mu \rangle$$
$$\geq 0,$$

since $f_1$ is the projection for $\langle \cdot, \cdot \rangle$ of $\mu$ onto the convex set $\mathcal{F}$. This yields the first claim. $\qquad\square$

*Proof of Theorem 4.* From the definition of the range of the outcome and of the regression functions, $o \mapsto \sqrt{M}$ is an envelope for $\mathcal{F}$ and $o \mapsto 4M$ is an envelope for $\ell(\mathcal{F})$. From Lemma 7, and the fact that $\ell$ is $4\sqrt{M}$-equiLipschitz w.r.t. its first argument,

$$N_{[]}(4M\epsilon, \ell(\mathcal{F}), \|\cdot\|_{2, g^*}) \lesssim N_{[]}(\sqrt{M}\epsilon, \mathcal{F}, \|\cdot\|_{2, g^*}) \lesssim \epsilon^{-p},$$

where the last inequality follows from the fact that Assumption 6 holds for $\mathcal{F}$ with envelope $o \mapsto \sqrt{M}$. Therefore, Assumption 2 holds for envelope $\Lambda : o \mapsto 4M$. In addition, for this envelop definition, Assumption 3 holds with $\rho_0 = b_0 = 1$. Finally, from lemma Lemma 2,

$$\|\ell(f) - \ell(f_1)\|_{2, g^*} \leq 4\sqrt{M}(R^*(f) - R^*(f_1))^{1/2}$$
$$= 2(4M) \left( \frac{R^*(f) - R^*(f_1)}{4M} \right)^{\frac{1}{2}},$$

that is Assumption 4 holds. Theorem 4 then follows directly by instantiating Theorem 2 and Theorem 3, respectively in the case $p > 2$ and in the case $p \in (0, 2)$, with $\Lambda : o \mapsto 4M$, $\alpha = 1$, $b_0 = \rho_0 = 1$. $\qquad\square$

## D   Proof of the results on policy learning using ISWERM

*Proof of Theorem 5 and Theorem 6.* Note that since the outcome has range $[-M, M]$, $\ell$ is $M$-equiLipschitz w.r.t. its first argument. Therefore, from Lemma 7 and the fact that $\mathcal{F}$ satisfies Assumption 6 with envelope constant equal to 1, Assumption 2 holds with envelope $\Lambda : o \mapsto M$.

Furthermore, Assumption 3 holds for $b_0 = \rho_0 = 1$. Therefore, instantiating Theorem 2 with $\Lambda = M$, $\rho_0 = b_0 = 1$ yields Theorem 5.

Under realizability and Assumption 7, Lemma 3, gives us that Assumption 4 holds for $\alpha = \nu/(\nu + 1)$. Theorem 6 follows by instantiating Theorem 6 with $\alpha = \nu/(\nu + 1)$, $b_0 = \rho_0 = 1$. $\qquad\square$

# E Technical lemmas

## E.1 Long version of lemma 4 in [10]

We restate here under our notation the full version of lemma 4 in [10], of which we gave a short version under the form of lemma 1.

**Lemma 7** (Lemma 4 in [10], long version). *Let $\ell : \mathcal{F} \times \mathcal{O} \to \mathbb{R}$. Suppose that there exists $\widetilde{\ell} : \mathbb{R} \times \mathcal{O} \to \mathbb{R}$ such that*

- *it holds that $\forall f : \mathcal{O} \to \mathbb{R}, o \in \mathcal{O}, \ell(f, o) = \widetilde{\ell}(f(o), o)$,*

- *$\widetilde{\ell}$ is L-equiLipschitz w.r.t. its first argument, that is,*

$$|\widetilde{\ell}(z_2, o) - \widetilde{\ell}(z_1, o)| \leq L|z_1 - z_2|, \forall o \in \mathcal{O}, z_1, z_2 \in \mathbb{R}$$

- *for every $o \in \mathcal{O}$, $z \mapsto \widetilde{\ell}(z, o)$ is unimodal.*

*Then, for any measure $\mu$ on $\mathcal{O}$, any $p \geq 1$, and $\epsilon > 0$, it holds that*

$$N_{[]}(L\epsilon, \ell(\mathcal{F}), \| \cdot \|_{\mu,p}) \leq N_{[]}(\epsilon, \mathcal{F}, \| \cdot \|_{\mu,p}).$$

## E.2 Proof of the variance bound under margin condition

*Proof of Lemma 3.* By assumption there exists $f_1 \in \mathcal{F}$ such that $R^*(f_1) = \mathbb{E}_{p_X} \mu^*(X)$. Applying Assumption 7 with $u = 0$ shows that we necessarily have $|\mathrm{argmin}_{a \in \mathcal{A}} \mu(X, a)| = 1$ almost surely. Therefore, almost surely, $f_1(X, a^*(X)) = 1$ and $f_1(X, a) = 0$ for $a \neq a^*(X)$.

Now fix any $f \in \mathcal{F}$. Given $X$, let $A \in \mathcal{A}$ be random variable draw from $f(X, \cdot)$. We will henceforth denote expectations and probabilities as wrt $(X, A) \sim p_X \times f$. For brevity we will also denote $A^* = a^*(X)$. Note that

$$\|\ell(f, \cdot) - \ell(f_1, \cdot)\|_{2,g^*}^2 \leq M^2 \mathbb{P}(A^* \neq A)$$

and that

$$\|\Lambda\|_{2,g^*}^2 \left( \frac{R^*(f) - R^*(f_1)}{\|\Lambda\|_{2,g^*}} \right)^\alpha = M^2 (\mathbb{E}[\mu(X, A) - \mu(X, A^*)] / M)^{\nu/(\nu+1)}.$$

Denoting $\Delta = \min_{a \in \mathcal{A} \setminus \{a^*(X)\}} \mu(X, a) - \mu^*(X)$, Assumption 7 says that for some $\kappa > 0$ we have $\mathbb{P}(\Delta \leq u) \leq (\kappa u / M)^\nu$, where $1^\infty = 1$ and $x^\infty = 0$ for $x \in [0, 1)$.

Fix $u > 0$. Then

$$\begin{aligned}
\mathbb{E}[\mu(X, A) - \mu(X, A^*)] &= \mathbb{E}[(\mu(X, A) - \mu(X, A^*))\mathbf{1}(A \neq A^*)] \\
&\geq \mathbb{E}[(\mu(X, A) - \mu(X, A^*))\mathbf{1}(A \neq A^*, \Delta > u)] \\
&\geq u\mathbb{P}(A \neq A^*, \Delta > u) \\
&= u(\mathbb{P}(A \neq A^*) - \mathbb{P}(A \neq A^*, \Delta \leq u)) \\
&\geq u(\mathbb{P}(A \neq A^*) - \mathbb{P}(\Delta \leq u)) \\
&\geq u(\mathbb{P}(A \neq A^*) - (\kappa u/M)^\nu).
\end{aligned}$$

Set $u = ((\nu + 1)\kappa/M)^{-1/\nu}\mathbb{P}(A \neq A^*)^{1/\nu}$ and obtain

$$\mathbb{E}[\mu(X, A) - \mu(X, A^*)] \geq \nu(\nu + 1)^{-(\nu+1)/\nu}(\kappa/M)^{-1}\mathbb{P}(A \neq A^*)^{(\nu+1)/\nu},$$

whence

$$\mathbb{P}(A \neq A^*) \leq \nu^{-\nu/(\nu+1)}(\nu + 1)((\kappa/M)\mathbb{E}[\mu(X, A) - \mu(X, A^*)])^{\nu/(\nu+1)}.$$

We conclude that

$$\|\ell(f, \cdot) - \ell(f_1, \cdot)\|_{2,g^*}^2 \lesssim M^2 (\mathbb{E}[\mu(X, A) - \mu(X, A^*)] / M)^{\nu/(\nu+1)}$$

as desired. □

# F Additional Details and Results for the Empirical Investigation

Here we provide additional details and results for Section 6.

## F.1 Contextual Bandit Data from Multi-Class Classification Datasets

To construct our data, we turn $K$-class classification tasks into a $K$-armed contextual bandit problems [15, 17, 51], which has the benefits of reproducibility using public datasets and being able to make uncontroversial comparisons using actual ground truth data with counterfactuals. We use the public OpenML Curated Classification benchmarking suite 2018 (OpenML-CC18; BSD 3-Clause license) [11], which has datasets that vary in domain, number of observations, number of classes and number of features. Among these, we select the classification datasets which have less than 60 features. This results in 51 classification datasets from OpenML-CC18 used for evaluation. Table 1 summarizes the characteristics of the 51 OpenML datasets used.

| Samples | Count | | Classes | Count | | Features | Count |
|---|---|---|---|---|---|---|---|
| $< 1000$ | 16 | | $= 2$ | 30 | | $\geq 2$ and $< 10$ | 14 |
| $\geq 1000$ and $< 10000$ | 25 | | $> 2$ and $< 10$ | 15 | | $\geq 10$ and $< 30$ | 22 |
| $\geq 10000$ | 10 | | $\geq 10$ | 6 | | $\geq 30$ and $\leq 60$ | 14 |

Table 1: Characteristics of the 51 OpenML-CC18 datasets used for evaluation.

Each dataset is a collection of pairs of covariates $X$ and labels $L \in \{1, \ldots, K\}$. We transform each dataset to the contextual bandit problem as follows. At each round, we draw $X_t, L_t$ uniformly at random with replacement from the dataset. We reveal the context $X_t$ to the agent, and given an arm pull $A_t$, we draw and return the reward $Y_t \sim \mathcal{N}(\mathbf{1}\{A_t = L_t\}, 1)$. To generate our data, we set $T = 100000$ and use the following $\epsilon$-greedy procedure. We pull arms uniformly at random until each arm has been pulled at least once. Then at each subsequent round $t$, we fit $\widehat{\mu}_{t-1}$ using the data up to that time. Specifically, for each $a$, we take the data $\{(X_s, Y_s) : 1 \leq s \leq t-1, A_s = a\}$ and pass it to a regression algorithm in order to construct $\widehat{\mu}_{t-1}(\cdot, a)$. In Section 6, we presented results where we use `sklearn`'s `LinearRegression` to fit $\widehat{\mu}_{t-1}(\cdot, a)$ (using `sklearn` defaults). In Appendix F.2, we repeat the experiments where we instead use `sklearn`'s `DecisionTreeRegressor` (using `sklearn` defaults). We set $\tilde{A}_t(x) = \operatorname{argmax}_{a=1,\ldots,K} \widehat{\mu}_{t-1}(a, x)$ and $\epsilon_t = t^{-1/3}$. We then let $g_t(a \mid x) = \epsilon_t/K$ for $a \neq \tilde{A}_t(x)$ and $g_t(\tilde{A}_t(x) \mid x) = 1 - \epsilon_t + \epsilon_t/K$. That is, with probability $\epsilon_t$ we pull a random arm, and otherwise we pull $\tilde{A}_t(X_t)$.

## F.2 Additional Results

In Section 6, we presented results where we use a linear-contextual $\epsilon$-greedy bandit algorithm to collect the data. Here, we repeat our experiments when the data are instead collected by a tree-contextual $\epsilon$-greedy bandit algorithm, as described in Appendix F.1 above. The results are shown in Fig. 2. The conclusions are generally the same: ISWERM compares favorably for fitting linear models, while all methods perform similarly for fitting tree models.

## F.3 Code and Execution Details

The IPython notebook to reproduce the experimental results of the main paper and the appendix is included as an attachment in the Supplemental Material. One needs to obtain an OpenML API key to run this code (instructions can be found at https://docs.openml.org/Python-guide/) and replace the string `'YOURKEY'` in `summarize_openmlcc18()` and in `download_openmlcc18()` functions with it. After that, if the notebook is executed as is, it reproduces Figure 1 (38h 26min on a single Intel Xeon machine with 32 physical cores/64 CPUs). Changing variable `bandit_model` from `'linear'` to `'tree'` reproduces Figure 2 (56h 45min on a single Intel Xeon machine with 32 physical cores/64 CPUs).


Figure 2: Comparison of weighted regression run on contextual-bandit-collected data. Each dot is one of 51 OpenML-CC18 datasets. Lines denote $\pm 1$ standard error. Dots are blue when ISWERM is clearly better, red when clearly worse, and black when indistinguishable within one standard error.

- (a) Do the main claims made in the abstract and introduction accurately reflect the paper's contributions and scope? [Yes]
- (b) Did you describe the limitations of your work? [Yes]
- (c) Did you discuss any potential negative societal impacts of your work? [Yes]
- (d) Have you read the ethics review guidelines and ensured that your paper conforms to them? [Yes]

2. If you are including theoretical results...

- (a) Did you state the full set of assumptions of all theoretical results? [Yes]
- (b) Did you include complete proofs of all theoretical results? [Yes]

3. If you ran experiments...

- (a) Did you include the code, data, and instructions needed to reproduce the main experimental results (either in the supplemental material or as a URL)? [Yes] In the supplemental material with specifics in Section F.3 of the supplemental material.
- (b) Did you specify all the training details (*e.g.*, data splits, hyperparameters, how they were chosen)? [Yes] In Section 6
- (c) Did you report error bars (*e.g.*, with respect to the random seed after running experiments multiple times)? [Yes] In both Figure 1 and Figure 2
- (d) Did you include the total amount of compute and the type of resources used (*e.g.*, type of GPUs, internal cluster, or cloud provider)? [Yes] In Section F.3 of supplemental material.

4. If you are using existing assets (*e.g.*, code, data, models) or curating/releasing new assets...

- (a) If your work uses existing assets, did you cite the creators? [Yes] In Section F.1 of the Supplemental Material.
- (b) Did you mention the license of the assets? [Yes] In Section F.1 of the Supplemental Material.

(c) Did you include any new assets either in the supplemental material or as a URL? [N/A]

(d) Did you discuss whether and how consent was obtained from people whose data you're using/curating? [N/A]

(e) Did you discuss whether the data you are using/curating contains personally identifiable information or offensive content? [N/A]

5. If you used crowdsourcing or conducted research with human subjects...

(a) Did you include the full text of instructions given to participants and screenshots, if applicable? [N/A]

(b) Did you describe any potential participant risks, with links to Institutional Review Board (IRB) approvals, if applicable? [N/A]

(c) Did you include the estimated hourly wage paid to participants and the total amount spent on participant compensation? [N/A]