# OpenReview forum: "Risk Minimization from Adaptively Collected Data: Guarantees for Supervised and Policy Learning"
_NeurIPS.cc/2021/Conference — NeurIPS 2021 Poster_

### Official Review · Reviewer_5gHo · 2021-07-14

**Rating:** 5
**Confidence:** 2

**Summary:**

This work studies ERM for adaptively collected data with a generic importance sampling weighted risk. The main contribution is theoretical, where the authors claim to provide first-of-their-kind generalization guarantees and fast convergence rates. Technically, the results rely on a new maximal inequality the authors derive by leveraging the importance sampling structure. A short experiment is also provided to validate the theory.

**Limitations And Societal Impact:**

Yes

**Main Review:**

The paper seems to contain new theoretical insights which might be interesting. However, the current version is technically dense and not very reader friendly. I think it is very hard to appreciate the results (note: this could be due to my lack of relevant background, so I would leave the experts to comment on the technical depth). The paper has extensive citations on prior work and it seems to require a fairly broad and detailed understanding of the literature. Often, the results are missing enough intuition/explanation. It is not easy to understand that beyond the mathematical equations, what the theory would imply and how the general audience should interpret it. Some of the assumptions are assumed without further explanation and hence, it is not clear whether they are assumed solely for the technical proof or there are deep connections to the problem. Overall,  I would suggest to revise the paper to improve the readability. Also, there are some notable typos (e.g., lines 145 and 151) that can be fixed.

**Time Spent Reviewing:**

2h

---

> ### Author Response · Authors · 2021-08-10
> **Authors' Response to Reviewer 5gHo**
>
> Thank you for taking the time to review our paper. We were glad to read that you found our results potentially interesting. We are sorry you feel the paper is too dense. While we worked very hard to communicate a complex but important result in a clear and concise way, we can certainly improve it further, and your feedback is immensely helpful in that.
>
> You mentioned that we have many citations on prior work that undermines clarity and requires a detailed understanding of the literature. We respectfully disagree. In fact, our point with discussing these citations is to provide the appropriate contextualization of our result in the literature and to help the reader understand the appropriate context *without* having to know the particular details of all these papers. Not having these references and discussions would only hurt clarity in the sense that the context of our results in the literature would be harder for a reader to understand.
>
> Regarding the assumptions, we can certainly add some clarity by better explaining their intent here and in the paper. Namely, Assumption 1 is an assumption about the exploration rate of the bandit algorithm and it is our most crucial and fundamental assumption, which we will emphasize. Assumptions 2-5 are purely technical devices in order to state Theorems 2 and 3, and they are established in specific instantiations of ISWERM such as regression or policy learning (note they do not appear in Theorem 4-6); we can definitely make this clearer. For example, Assumption 6 provides a simple way to ensure Assumption 2. Finally, Assumption 7 is about the amount of noise near the decision boundary; many references are given in the paragraph following its statement about it.
>
> Thank you also for pointing out some typos. It is very helpful and indicates a close reading. We greatly appreciate it, and we will fix them.

---

### Official Review · Reviewer_ZSG9 · 2021-07-14

**Rating:** 7
**Confidence:** 3

**Summary:**

This paper develops theoretical guarantees for importance-weighted empirical risk minimization on adaptively collected data. They characterize the risk of their method compared to the population risk minimizer in the form of a high probability bound. This type of bound can be used for guaranteeing risk minimization for policy learning on adaptively collected data.

**Limitations And Societal Impact:**

See discussion in main review above.

**Main Review:**

The bound presented in this paper is an important result for how statistical methods for i.i.d. data can be modified for adaptively collected data, which includes all types of data collected with bandit algorithms. It is the first bound of its kind for adaptively collected data.

I think the authors should discuss the following limitation in the discussion section: It is not clear that importance weighted empirical risk minimization is necessarily better than unweighted empirical risk minimization (or ERM with different weights) on adaptively collected data. This agrees with Figure 1 (c) where unweighted ERM empirically outperforms importance weighted ERM.

I found significant portions of the text difficult to read, e.g., some of the notation is not explicitly defined in the text.
-	Line 140: Did not define what script O is yet.
-	Line 147: I got very confused thinking that k denoted an exponent. Perhaps put parentheses around k.
-	Line 282: What is the mu function here?

From reading the main text of the paper, it is very difficult to interpret the simulation results. It would be useful to provide some more information in the main text.
-	What is g^* in your simulations?
-	Explicitly say what the axes represent in Figure 1 in caption.

**Time Spent Reviewing:**

3 hours

---

> ### Author Response · Authors · 2021-08-10
> **Authors' Response to Reviewer ZSG9**
>
> Thank you for your reviewer. We heartened to read that you appreciated the importance of the paper.
>
> Your point about ISWERM vs ERM is an important but nuanced one, and we’ll definitely discuss this more explicitly, and in fact exhibiting this point was exactly the intent of our experiments, which we will make clearer. Our focus is on *risk minimization* (per our title), that is, finding the hypothesis with the smallest population risk in a class, and in general ERM simply cannot obtain such best-in-class guarantees in our setting. Think about an iid setting with covariate shift (e.g., the treated population vs the population at large in causal inference): ERM would target the wrong population risk (on the treated) and cannot obtain optimal risk (on everyone), whereas ISWERM would. If, however, we consider a well-specified model, then being optimal on any one distribution means being optimal on any other (subject to some overlap) because we are simply optimal at every single X, so ERM would be fine, but that is different from *risk minimization* (aka model agnostic ML). In practice, the closer we are to unrestricted models, the less impact weighting has, exactly as Fig 1(c) shows. To show this was exactly our intent in the experiments -- that this behavior persists as expected also in the adaptive settings, as our theory establishes -- we will make this intent clearer. We will also cite the reference Rakhlin et al. “Empirical entropy, minimax regret and minimax risk”, which studies theoretically this disparity between optimally-achievable risk and regret (in the iid setting without covariate shift).
>
> Thank you also for helpful pointing out places where we can improve clarity; we’ll use this to improve our paper. Below we respond:
> - Line 140: $\mathcal O$ was defined on line 43; we’ll remind the reader here for clarity.
> - Line 147: Good point; we’ll make these into $\square^{(k)}$.
> - Line 282: $\mu$ was defined on line 58. Admittedly it’s inside an example, which is confusing, so we’ll take it out and define it clearly separately.
> - $g^*$ in the simulations is $g^*=1/K$ as noted on line 317 (which is essentially the same as $g^*=1$, but we wanted it to be a proper policy so that the Monte-Carlo evaluation we describe makes sense). We will make this more explicit.
> - Great suggestion re Fig 1 caption: it is MSE, as noted in the text but indeed sadly not in the caption.

---

> > ### Comment · Reviewer_ZSG9 · 2021-08-25
> > **Response**
> >
> > Thank you for your clarifications. After reading what the other reviewers have said, I have lowered my score slightly. My concerns are with (1) the overall clarity of the presentation, which all other reviewers have also brought up, and intuition / proof sketch for the results and (2) the comments brought up by reviewer q5vX regarding precise discussion of relationship to prior work, particularly Zhan et al.

---

### Official Review · Reviewer_P7T1 · 2021-07-16

**Rating:** 7
**Confidence:** 2

**Summary:**

In this paper, Importance Sampling Weighted ERM (ISWERM) is studied for the use of adaptively collected data to minimise the average of a loss function over an hypothesis class to get generalisation guarantees and fast convergence rates. The main result to achieve this is a novel maximal inequality for importance sampling weighted empirical processes. The authors then apply this result to obtain slow rates for certain entropy condition and fast rates under variance bounds; and they apply this to regression and policy learning. The authors conclude with an empirical evaluation of ISWERM against many variations on this approach, using contextual bandit data.

**Limitations And Societal Impact:**

The authors did address some concerns, but maybe the limitations could be addressed a bit more. On the other hand, they show that the results apply to some commonly used settings and methods.

**Main Review:**

The paper is well-written and pleasant to read. The structure is very clear. There is a thorough comparison with related literature.

The main theoretical contribution is a maximal inequality for importance sampling weighted sequential empirical processes (theorem 1). I'm not an expert in this area, so I am unable to judge the depth of this result. It seems a sound result. The result can be used to obtain generic guarantees for ISWERM: slow rates under certain entropy conditions, and, as to be expected, fast rates in the presence of variance bounds. ISWERM is instantiated to regression and policy learning, which shows applicability to important problems for which ERM is commonly used (for iid data), so the scope of these results is satisfying. Also, the comparison (remark 3 and 4) with existing literature is very nice, as is the empirical evaluation - where ISWERM is compared to other weighting methods from the literature and performed with ridge, lasso regression and cart - also very useful methods, so: good choice.

It the other reviewers have a better estimation of the importance/impact of the results, I'm happy to lower or higher my score.

Some minor remarks:
line 80: bilbioraphies -> bibliographies

Van der Laan, Van de Geer, Van Handel, Van der Vaart:
Like many Dutch names, these surname consists of two or three words: 'Van de Geer'. The article 'Van' is capitalised, except when directly preceded by a given name or initials. Although 'Van de' always precedes 'Geer', in an alphabetical list, this name should be listed according to 'Geer' only. Inline it should thus look like: "Van de Geer [55]" and in the bibliography it should be sorted according to the letter G and look like: "Sara van de Geer ...".

**Time Spent Reviewing:**

3

---

> ### Author Response · Authors · 2021-08-10
> **Authors' Response to Reviewer P7T1**
>
> Thank you for taking the time to review our paper. We are glad you appreciated both the value of the results and the clarity of exposition and comparison. We worked hard to communicate our results clearly, correctly, and concisely.
>
> Thanks for the note on the Dutch names. Unfortunately, natbib requires some pretty hacky ways to get around this, but we’ll implement the hacks so that this is done right in respect of Dutch heritage.
>
> And thanks for catching the typo: we will fix it.

---

### Official Review · Reviewer_kdLu · 2021-07-16

**Rating:** 6
**Confidence:** 3

**Summary:**

This paper considers the problem of empirical risk minimization when the data are adaptively collected and are thus non i.i.d. The data can have been sampled from a sequential contextual bandit procedure for instance. Theoretical guarantees for classical empirical risk minimization fail in this case. This paper analyses the empirical risk minimizer reweighed via importance sampling (ISWERM). For non-parametric classes of functions, the authors provide upper-bound on its excess risk. The latter is expressed in terms of a sequential bracketing entropy of the class of function. When the data is i.i.d., one retrieves the classical slow rate of convergence for non-parametric risk minimization. The authors also show faster rates under additional assumptions (variance bound, square loss, margin condition). In the case of policy learning with epsilon-greedy as the sampling policy, the upper-bound matches the known lower bound closing thus an important gap. Finally, ISWERM is shown to empirically outperform standard unweighted ERM and other weighted versions of ERM.

**Limitations And Societal Impact:**

Yes

**Main Review:**

To my opinion, this paper brings a significant contribution to the community. ERM is widely used and having simple versions of it that come with theoretical guarantees when the data is non i.i.d. but adaptively sampled is surely interesting. Furthermore, the analysis seems nontrivial and requires sophisticated technical tools from empirical process maximization and I think that matching the lower-bound for policy learning was a challenge.


On the other hand, the writing of the document is sometimes very technical with many notations and requires a lot of concentration to be understood. I wonder if a preliminary section that highlights the results and provides sketches of proofs for finite classes of functions (or parametric, so that bracketing entropy is not necessary) might not help in understanding the proof and the results. It could be also interesting to recall some classes of functions and losses where the ISWERM can be efficiently estimated.

Other comments:
1. As mentioned in the conclusion, I agree that it would surely be interesting to extend the analysis to variance reduction of IPS, such as doubly robust since IPS can be very unstable. If the analysis is surely technical and can be left for future work, I think that It could be of interest to add for instance a doubly robust version in the experiments? I am a bit surprised that standard IPS seems to outperform more sophisticated versions that were designed to be more stable such as ISFloorWERM (and others). What are the advantages of such methods then?
2. I would also find it interesting to analyze a clipped version of IPS when the weights get too large. For some bandit algorithms, gamma_t can grow very quickly. Clipping might be helpful in this case to reduce the variance.
3. Experiments with synthetic data that show the rates of convergence indeed obtained by ISWERM and ERM could also be of interest.
4. After Thm.1, I would emphasize that this bound is very close to what is known in probability theory as
Dudley’s entropy integral, a useful tool to upper bound the expectation of a centered stochastic process with subgaussian increments. And that it is often key to derive risk bounds on empirical risk minimizers in statistical learning.
5. What about the case p=2 in Thm. 2, 4,... Sometimes, the two cases do not match when p-> 2, is it normal?


Minor remarks and typos:
- section 1.2: sometimes $g^*(x|a)$ instead of $g^*(a|x)$
- l145: sequence sequence
- l146: I would define the norm $\|| \cdot \||_{2,g}$ there
- Ass 2: the notation of the paper is already heavy, I do not understand why you need to introduce the envelope function Λ when you always use $\||\Lambda\||_{2,g}$ which could be introduced by the existence of a positive number only.
- l233: withe
- l300: multi-calss

**Time Spent Reviewing:**

4

---

> ### Author Response · Authors · 2021-08-10
> **Authors' Response to Reviewer kdLu**
>
> We were heartened to hear you think “this paper brings a significant contribution to the community”. Thank you for taking the time to review the paper and for your feedback on how to improve the exposition. We worked hard to try and communicate our results clearly and concisely, and we appreciate your help in making it even better. Thank you for the suggestion for a preliminary sketch section; we will implement it.
>
> As to your “Other comments” in order:
> 1. While we think we know how to extend our results to DR-estimation-based ERM (it follows by applying case 2 of our Theorem 1 as long as for unit t we use outcome models fit only data up to t-1), we believe it is out of scope for this short paper as it is already technical and has many results. We will certainly mention the possibility of this extension and sketch the proof. A more complete investigation must be left to future work, however, for the chance of clearly communicating it.\
> Regarding outperforming methods designed to be stable, stabilized weights stabilize the variance of the estimator evaluating a single hypothesis so that the estimator satisfies the conditions of Martingale CLT and ensures asymptotic normality, it need not mean that optimizing such an objective over many hypotheses is more stable. We will clarify this. In some cases it may be even change the population target, such as the case of SqrtISWERM, which was studied in the context of ERM in Zhang et al. 2021 as a stabilization method. Although under well-specification, Zhang et al. show it enables inference on model parameters, if the model is misspecified, we need not obtain minimum risk over the hypothesis class by SqrtISWERM.
> 2. That does sound interesting and we’d be glad to mention it. Our conjecture, however, is that this cannot improve the bound, but it may perform great in practice.
> 3. Good idea. We’ll add log-log MSE plots with linear fits that show rates of convergence (i.e., $n$ to what power) in our regression experiments, so we can compare to the rates predicted by our theory.
> 4. We will definitely note this. Note, however, as noted in Remark 2, we use an adaptive chaining as pioneered by Ossiander [41]. That is a refinement of the classic chaining as used originally by Duddley. We will clarify this historical development.
> 5. As we note on line 192, the p=2 case behaves like the p>2 case with p->2 but with extra log factors. We omitted this case for brevity. This is normal as there is a breakpoint in the analysis between the finite entropy integral case (p<2) and infinite case (p>=2), and this appears in classic analyses of ERM. See, for example, Rakhlin et al. “Empirical entropy, minimax regret and minimax risk”, which we’ll cite as a reference.
>
> Finally, thank you for pointing out our typos. This indicates a close and concentrated reading for which we are very grateful. We will certainly fix these.
>
> Regarding $\vert\vert h\vert\vert_{2,g}=P_g(h^2)$, we should indeed define that on line 141. Regarding $\Lambda$, we do need an envelope, not a constant positive bound, and we need to refer to the L2 norm of a valid envelope; we propose to give $\|\Lambda\|_{2,g^*}$ a letter name, if that’s helpful.

---

> > ### Comment · Reviewer_kdLu · 2021-08-19
> > **Response to the authors**
> >
> > I thank the authors for their response to my review. While I am currently travelling (for the next 1.5 week) and in a very tight time schedule, I only give a high level response. I also read other reviews.
> >
> > Overall, my opinion of the paper did not change. I believe that on one hand, the technical contribution of the paper is significant enough, but on the other hand the clarity could be improved (though I acknowledge that this is not easy on this technical topic). I am happy to keep my score of 7 with the hope that the authors will do some effort (add sketches of proofs...) to improve the readibility of their results if the paper gets accepted.

---

> > > ### Comment · Reviewer_kdLu · 2021-08-31
> > > **Follow-up**
> > >
> > > After reading the other reviews in more detail, I think the paper, the authors, and the community would benefit from a fairly thorough rewrite of the paper to make its results more accessible. This seems to be difficult within the page limit required for Neurips and would be more in line with a major revision. So I finally lower my score to 6.

---

> > > > ### Author Response · Authors · 2021-08-31
> > > > **Rejoinder**
> > > >
> > > > We greatly appreciate your taking the time to read our paper closely and consider our responses and others' reviews, and we thank you for the very useful feedback. We respectfully disagree about the fit for NeurIPS and, while we sincerely appreciate the incredibly useful feedback pointing out at where can clarity be improved, we do not think addressing the concerns requires anything close to a rewrite. We have worked very hard to cleanly, concisely, correctly, and clearly communicate our results, which have direct implications to very important and timely topics such as causal learning. NeurIPS is the leading venue for the topic of causal ML, including theoretical contributions, and our work has the potential for significant impact in this space. We do believe we largely succeeded in being able to communicate our results convincingly within the length limits of NeurIPS. We recognize we can always improve and we will use the excellent feedback we got to further enhance clarity at the specific points kindly pointed out to us by you and your colleagues, but this does not involve anything close to a rewrite. We do hope you agree and will support our paper being accepted as we think it makes an important contribution to the area. Thank you again.

---

### Official Review · Reviewer_q5vX · 2021-08-01

**Rating:** 5
**Confidence:** 4

**Summary:**

The paper provides a nice technical contribution; the topic of the paper is relevant. I appreciate the technical contribution of the paper and some extensions of the empirical process theory. However, as a referee, I need to answer if the paper is interesting to the audience and whether it should be presented at the conference. Overall, it looks like an example paper where too much technical content shadows the paper's conceptual message. Also, it seems that once Theorem 1 is established, all applications, including fast rates, become very standard.

Here are my main comments:

1) The authors claim that their results capture the correct dependence on the exploration rate. Does it mean that previous results on weighted ERM could not achieve this? Or is it only related to the recent work Zhan et al. ? It seems that van de Geer (who used sequential bracketing inequalities) did not study weighted ERM.

2) A few remarks in the main text look too specialized/technical. For example, Remarks 2, 3, 4 occupy a significant part of the main text but discuss relatively subtle technical details appearing in the previous papers. Moreover, the assumptions and models in these previous papers are different. In particular, I found a claim that the authors match the lower bound of Zhan et al. 21 closing the existing gap quite confusing: the lower bound of Zhan et al. 21 is in a different setup involving the Natarajan dimension of the policy class and so on (while the authors prefer non-parametric (even non-Donsker) functional classes).

3) The results are based on the bracketing entropies. It is quite restrictive for parametric classes. In particular, for some "small" classes like VC-subgraph/Natarajan classes, bracketing entropies can behave very poorly and thus, standard parametric rates cannot be recovered using this technique. This is also relevant when comparing with previous "parametric" papers.

4) It seems that fundamental results on sequential empirical processes (Theorem 1) are more suitable for specialized journals rather than ML conferences, where the technical aspects of the proofs cannot be properly verified/appreciated. As I mentioned, apart from this contribution, other technical parts of the paper are standard. If Theorem 1 is of such an interest, it is better to make a more detailed comparison with previous results ( at least write van de Geer's bound explicitly).

Overall, I think it is a good piece of research, but I could not recommend it for acceptance in the present form.

Technical comments.

1) What is the definition of \| \|_{2, g*} norm first appearing in line 146?
2) Lines after line 621 - missing brackets.
3) Typo after line 628.
4) How is the proof of Theorem 6 based on Theorem 6 itself in line 661?

**Limitations And Societal Impact:**

The authors have addressed the limitations and potential negative societal impact of their work.

**Main Review:**

The paper is a nice technical contribution and the topic of the paper is relevant. I appreciate the technical contribution of the paper and some extensions of the empirical process theory. However, as a referee, I need to answer if the paper is interesting to the audience and should be presented at the conference. Overall, it looks like an example paper where too much technical content shadows the paper's conceptual message. Once Theorem 1 is established, all applications, including fast rates become very standard.


**Time Spent Reviewing:**

6

---

> ### Author Response · Authors · 2021-08-10
> **Authors' Response to Reviewer q5vX**
>
> Thank you for taking the time to review our paper. We were glad to read that you found it a nice technical contribution on a relevant topic. We are sorry you feel the paper is too dense. While we worked very hard to communicate a complex but important result in a clear and concise way, we can certainly improve it further, and your feedback is immensely helpful in that.
>
> Let us answer your “main comments” in order:
> 1. You are correct that van de Geer did not study ISWERM; indeed, to get good dependence on gamma would require our modification of her maximal inequality. Her intent is simply different; not at all wrong. We will make that clearer. Regarding what we meant regarding correct dependence, we mean to refer to the fact that the dependence is the same as in the non-adaptive case where we only see explored data so we do not lose much due to adaptiveness (indeed, Zhan et al.'s lower bound arises by exhibiting a non-adaptive instance). We will make this much clearer and specific.
> 2. While admittedly technical, Remark 2 crucially points out the main difference in our main technical method and why we need it in order to establish the new subsequent guarantees on ISWERM, which itself is practical and commonplace. In Remarks 3 and 4, we only mean to provide a full context and give proper credit to other works, while pointing out an interesting open problem regarding dealing with adaptive data under covering entropy. You are absolutely right that our setting is slightly different than Zhan et al., and we do point out in Remark 3 that “the two complexity measures are simply different”. But, as we write, many classes, including trees and linear, satisfy *both* complexity measures and therefore in these cases we *do* match. We’ll make it clearer in what technically specific sense the bounds match. Also, it appears we made a small mistake in Remark 2 in talking about Hamming covering and not Natarajan dimension as appears in Zhan et al.’s lower bound (Natarajan gives a bound on the Hamming covering, which they use in their upper bound). Nonetheless, the same classes (trees and linear) still have both finite Natarajan and p<2 bracketing entropy, so we can still make a rigorous comparison of the lower and upper bounds for those classes. We will make this more explicit.
> 3. While it is true that in general VC and Natarajan dimensions do not imply bracketing entropies, parametric classes *do* have Donsker-like bracketing entropies (p>0 arbitrarily small because the entropy is logarithmic), as we mention in lines 204--214 (see Theorem 2.7.11 in van der Vaart and Wellner). Thus, we *do* achieve rates arbitrarily close to parametric (e.g., 1/n fast rates for regression). We will also make clear that, in fact, similar to the p=2 case (see line 192), the logarithmic entropy case behaves the same as taking p->0 in all of our bounds, with some additional logarithmic terms, so that we do in fact obtain $\tilde O(1/n)$ rates in the parametric case.
> 4. We respectfully strongly disagree. NeurIPS is the flagship ML conference and our results uniquely shed light on an important, commonplace ML method. We worked hard to present our results in a clear and concise manner and we are certain our work will find an interested audience in the participants of NeurIPS.
>
> Regarding your “Technical comments”:
> 1. Thanks for catching! It seems we forgot to define $||h||^2_{2,g}=P_g(h^2)$ for any $g$, that is, the L2 norm under $P_g$. That should be defined on line 141 after defining $P_g$.
> 4. We could not find the issue you reference in our text. Probably there is some typo due to a wrong \ref. We’d be glad to fix it if you can help us find it.

---

### Decision · Program_Chairs · 2021-09-28

**Decision:**

Accept (Poster)

**Comment:**

This looks like a strong paper that makes significant contributions. Simultaneously, the current presentation is very technically dense; this was the conclusion of absolute experts and so is not to be taken lightly. Consequently, despite the strength of the results in this work, the results will have a tough time being appreciated even by general readers in learning theory (and definitely going outside of learning theory). Also, there is no space in the current version to add anything, and so it is implausible that the authors would be able to add in things like proof sketches without a major revision of the paper. On that note, the authors mentioned that they do not believe the paper needs a major rewrite (the reviewers and I disagree), and a major revision would require another round of reviewing in any case.

Therefore, and I admit the decision is not easy, I really do feel that this paper should not be accepted in the current form. My feeling is that a resubmit to a conference like ALT, which is more generous with space and would allow for considerable improvements to the narrative (plus proof sketches, etc.) would be in service to the authors by letting their work be better appreciated by the field. On that note, I advise the authors to consider the reviewer feedback about significantly changing the presentation in their paper. It’s their choice whether or not to rewrite the paper, but to get it accepted somewhere, I (and evidently many reviewers) really think this is needed.

In the rest of this meta-review, I highlight some of the positive aspects that came out in the reviews, as well as some specific negative aspects.

On the positive side, the bound presented in this paper is the first of its kind for adaptively collected data. There certainly is novelty here, and the result is significant. The analysis is highly non-trivial (i.e., quite sophisticated). Based on the author response to some of the reviews and the paper itself, there also is interesting follow-up work.

On the negative side, the paper could really use more intuition (by way of informal statements) and proof sketches. However, there simply isn't space in the current form of the paper; it’s incredibly dense in terms of filling the space as it is. The authors disagree about reducing the lengthy, highly technical comparisons with previous papers (they seem unwilling to do a serious rewrite), even though several reviewers mentioned that this would be a good idea. Especially the fundamental result Theorem 1 should be given more justice/attention in the paper. In addition, one technical weakness is that a continuing concern about the comparison with the VC/Natarajan setup of the lower bound of Zhan et al. (when the authors claim that they have matched the lower bound). It sounds like the authors simply need to technically qualify in what way they match, and so I hope they clarify this in any future version.

**Consistency Experiment:**

NeurIPS has a long history of experimentation. In 2014, NeurIPS ran an experiment in which 10% of submissions were reviewed by two independent committees to quantify the randomness in the review process. This year, we repeated a variant of this experiment to see how the quality of the review process has changed over time.  This paper was part of the experiment and was therefore assigned to two committees (consisting of reviewers, an Area Chair, and a Senior Area Chair) that reached independent decisions.  If both committees made the same recommendation, this recommendation was followed. If a single committee recommended acceptance, the paper was accepted (with the exception of a few cases in which the other committee identified what we considered a fatal flaw, e.g., an error in a key result).

This copy’s committee reached the following decision: **Reject**

The other committee assigned to the paper recommended **Accept (Poster)**.  You can find the other set of reviews, along with any follow up discussion with the authors here:
https://openreview.net/forum?id=6KcBgHQz3sJ